# Multimodal cell-free DNA whole-genome TAPS is sensitive and reveals specific cancer signals

Dimitrios V. Vavoulis [1,2] ✉, Anthony Cutts[1], Nishita Thota[3], Jordan Brown [3], Robert Sugar[3], Antonio Rueda[3], Arman Ardalan[1], Kieran Howard [1], Flavia Matos Santo [1], Thippesh Sannasiddappa[3], Bronwen Miller[3], Stephen Ash[4], Yibin Liu [5,6], Chun-Xiao Song [4,7], Brian D. Nicholson [8], Helene Dreau[1], Carolyn Tregidgo[3] & Anna Schuh [1] ✉

The analysis of circulating tumour DNA (ctDNA) through minimally invasive liquid biopsies is promising for early multi-cancer detection and monitoring minimal residual disease. Most existing methods focus on targeted deep sequencing, but few integrate multiple data modalities. Here, we develop a methodology for ctDNA detection using deep (80x) whole-genome TET-Assisted Pyridine Borane Sequencing (TAPS), a less destructive approach than bisulphite sequencing, which permits the simultaneous analysis of genomic and methylomic data. We conduct a diagnostic accuracy study across multiple cancer types in symptomatic patients, achieving 94.9% sensitivity and 88.8% specificity. Matched tumour biopsies are used for validation, not for guiding the analysis, imitating an early detection scenario. Furthermore, in silico validation demonstrates strong discrimination (86% AUC) at ctDNA fractions as low as 0.7%. Additionally, we successfully track tumour burden and ctDNA shedding from precancerous lesions post-treatment without requiring matched tumour biopsies. This pipeline is ready for further clinical evaluation to extend cancer screening and improve patient triage and monitoring.

Earlier cancer detection has the potential to improve patient outcomes[1]. Current screening programmes around the world are limited to specific cancers (cervical, breast, colorectal, lung, prostate) that together make up less than 30% of all cancer diagnoses. Uptake of screening, especially of invasive procedures, depends on acceptance in the population. Multi-cancer early detection (MCED) using minimally invasive liquid biopsies therefore holds great promise. However, it also poses challenges from the inherent false positive rate in asymptomatic individuals caused by the low prevalence of cancer even in enriched risk groups[2,3].

Targeting early cancer detection to high-risk individuals presenting to primary care with symptoms of cancer represents an alternative approach[4,5]. However, many symptoms have poor predictive value for cancer in primary care, where the tools for risk stratification and patient triage remain limited[6,7]. In the UK National Health Service (NHS) in 2020/2021, 7.0% of 2.07 million referrals of symptomatic patients for urgent cancer investigation from general practitioners (GPs) resulted in a cancer diagnosis, accounting for 55% of cancer diagnoses that year[8]. Most patients present to their primary care

[1]Oxford Molecular Diagnostics Centre, Department of Oncology, University of Oxford, Oxford, UK. [2]Biomedical Research Centre, Centre for Human Genetics, University of Oxford, Oxford, UK. [3]Exact Sciences Innovation LTD, The Sherard Bldg, Edmund Halley Rd, Littlemore, Oxford, UK. [4]Ludwig Institute for Cancer Research, Nuffield Department of Medicine, University of Oxford, Oxford, UK. [5]College of Chemistry and Molecular Sciences, Wuhan University, Wuhan, China. [6]Taikang Centre for Life and Medical Sciences, Wuhan University, Wuhan, China. [7]Target Discovery Institute, Nuffield Department of Medicine, University of Oxford, Oxford, UK. [8]Nuffield Department of Primary Care Health Sciences, University of Oxford, Oxford, UK. ✉e-mail: dimitris.vavoulis@oncology.ox.ac.uk; anna.schuh@oncology.ox.ac.uk

provider with non-specific symptoms that could be caused by many cancer types and non-cancer diagnoses making the sequencing of investigations a challenge. Specialised services to investigate cancer across multiple sites in patients with non-specific symptoms have been introduced in some countries leading to a conversion rate of 8.1% across multiple cancer sites[9,10] or even higher (11–35%)[11]. Additional diagnostic technologies are urgently needed to assist primary care providers with the triage of symptomatic patients to reduce unnecessary referrals for cancer investigation[12–15].

The prospective observational cohort study SYMPLIFY[5] investigated recently the Galleri GRAIL MCED[16] that exploits cancer-specific targeted methylation signals, in symptomatic patients referred for cancer investigation in the NHS. The overall sensitivity and specificity were 66.3% (61.2–71.1%) and 98.4% (98.1–98.8%), respectively. Sensitivity increased with increasing age and cancer stage, from 24.2% (16.0–34.1%) in Stage I to 95.3% (88.5–98.7%) in Stage IV. Where a cancer signal was detected among cancer patients, the MCED test's prediction of the site of origin was correct in 85.2% (79.8–89%) of cases. SYMPLIFY showed the potential for MCEDs to direct investigation or referral towards cancers that guidelines would not, if based on symptoms alone.

In the NHS Genomic Medicine Service, and in many cancer centres across the world, whole genome sequencing (WGS) is already routinely available for genetic analysis of tumour samples. Expanding the WGS repertoire to indications that require ctDNA analysis would therefore be feasible. Most technologies for interrogating liquid biopsies deploy targeted deep sequencing of ctDNA and limit detection to the most common types of cancers and cancer-specific, acquired single nucleotide variants (SNVs). Some combine this information with targeted epigenetic analysis and protein markers to improve sensitivity across a wider range of tumour types[2,17]. Some studies have proposed interrogating liquid biopsies by shallow WGS for either cancer-specific, acquired copy number aberrations (CNAs)[18] or for specific ctDNA fragment attributes[19,20]. This alternative to targeted deep sequencing of SNVs supports the notion that breadth can replace depth[21]. Sparse data on few patients undergoing treatment has been published on integrating information from multiple modalities using deeper 30x-100x WGS[21–26]. Tissue-type specific methylation patterns are used to detect cancer signals and tissue of origin (TOO) in ctDNA and are historically derived from bisulphite sequencing, which is also employed in the Galleri assay. However, bisulphite treatment destroys up to 80% of available ctDNA reducing sensitivity significantly. It converts the 95% of unmethylated cytosines to thymines, destroying the genetic code for alignment and making SNV calling impossible.

TET-Assisted Pyridine Borane Sequencing (TAPS) is a base-level-resolution sequencing methodology for the detection of 5-methylcytosines and 5-hydroxy-methylcytosines[27–30]. Unlike bisulphite-sequencing, TAPS is a less destructive methodology, which employs a combination of TET enzyme with borane to exclusively convert the 5% of methylated cytosines, thus preserving the genetic code and opening the possibility of simultaneous methylome and genome analysis on the same sequencing data. Here, we aimed to distinguish true cancer signals in ctDNA from non-cancer noise using integrative multi-modal TAPS WGS in symptomatic cancer cases and non-cancer controls referred from primary care for cancer investigation.

## Results

We conducted a diagnostic accuracy study using a case-control design (Fig. 1 and Supplementary Data 1). Two-thirds of included patients were biological males and 71% were aged 60 years or older. For the cases with confirmed cancer, the median age was 67.5 years. They presented with a wide range of different specific or non-specific symptoms all representative of this patient group (Supplementary Data 1). In total, 5 (8.2%), 20 (32.8%), 35 (57.4%) and 1 (1.6%) patient were diagnosed with cancer stages 1, 2, 3 and 4, respectively. Just under two-thirds of patients had colorectal cancer. Follow-up data was available on all patients showing a median overall survival of 8.5 years for both early- and late-stage patients (Supplementary Fig. 1A). Patients with colorectal cancer had the longest median overall survival (>8.8 years), followed by oesophageal (>8.1 years), ovarian (8 years), renal (3.7 years) and pancreatic (2 years) cancer patients (Supplementary Data 1 and Supplementary Fig. 1B).

### Analysis of chromosomal alterations for the detection of ctDNA

Copy number aberrations (large losses or gains of chromosomal material) are considered a hallmark of cancer, manifesting early during tumorigenesis and persisting during subsequent stages of tumour evolution[31–33]. They may involve large regions of each chromosome, chromosome arms or whole chromosomes (aneuploidies). Such alterations manifest themselves in the data derived from a WGS assay as contiguous upward (gains) or downward (losses) deviations of the number of aligned reads from a baseline corresponding to the non-aberrant (diploid) state. The magnitude of each such deviation depends on the underlying copy number state of the genome at the locus of the aberration, and on the overall tumour fraction in the sequenced sample[32]. To the extent that these aberrations are present in the ctDNA captured and sequenced from the plasma of patients,

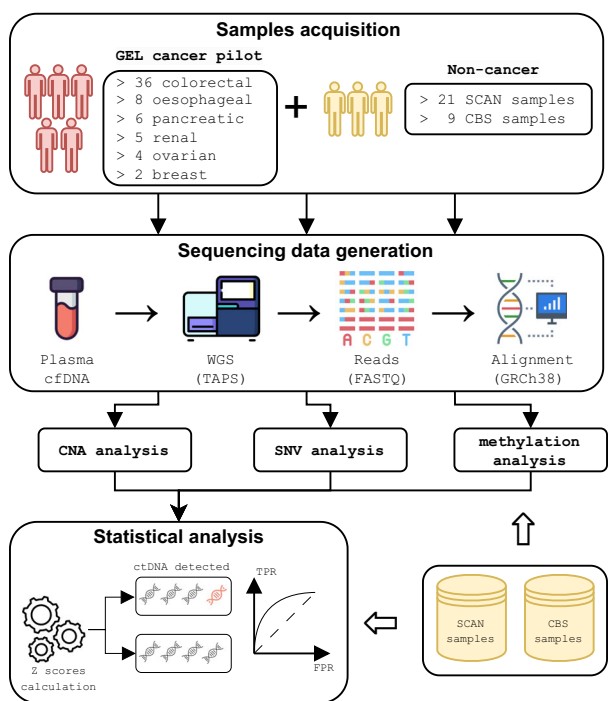

**Fig. 1 | Overview of the study.** We conducted a diagnostic accuracy study using a) cancer cases (GEL) from a cohort of symptomatic patients referred for urgent investigation for a possible gynaecological, lower GI, upper GI or renal cancer, and b) non-cancer controls with non-specific symptoms that might have been due to cancer from a rapid diagnostic centre (SCAN) or from Cambridge Bioscience (CBS). After collection of plasma samples, we conducted whole-genome sequencing at 80x or higher using TET-Assisted Pyridine Borane Sequencing (TAPS), aligned the generated reads against the human genome (GRCh38), and conducted analyses of copy number aberrations, methylation modifications, and somatic point mutations and indels, which included efficient denoising using the non-cancer SCAN controls. By integrating the analyses from all three data modalities, we generated sample-specific scores for the quantification of plasma ctDNA burden, which was used for cancer detection and post-treatment disease tracking. Matched tumour biopsies, if available, were used for validation, not for guiding the analysis. GEL Genomics England, SCAN Suspected Cancer Pathway, CBS Cambridge Biosciences Human Blood Products Supply Service.

they can be exploited for the non-invasive detection of almost all cancer types, due to the universal presence of such alternations in the pathogenesis of the disease.

For each cfDNA sample, we divided the genome in 1 kb-long non-overlapping bins and counted the number of proper, high-quality alignments that overlapped each bin, followed by a thorough filtering process for removing potential artefacts (see 'Methods'). Since next-generation sequencing and variable mapping can introduce characteristic biases, which alter the chromosomal representation of the original genomic DNA depending on local GC content and mappability[34], we further applied a statistical approach to simultaneously remove these two types of bias from the number of reads in each bin (see 'Methods'; Fig. 2Ai).

Further systematic bias can be introduced due to experimental (e.g., sample preparation) and biological factors (e.g., replication timing), which may hinder detection of subtle read depth differences across the genome in plasma samples with low cfDNA content. For this reason, we further applied a denoising process to remove any systematic errors (see 'Methods'). First, we characterised the background noise using principal component analysis on a panel of non-cancer plasma samples. These were obtained from 21 patients referred to a non-specific-symptoms cancer pathway (SCAN group; see 'Methods'), which were confirmed healthy or with a non-cancer diagnosis, and therefore appropriate to characterise any non-cancer-specific unwanted variation. This was followed by removing the background noise from each sample and, subsequently, applying a Savitzky-Golay smoothing filter, which further removes unwanted variation without distorting the underlying signal (Fig. 2Aii–iv).

To imitate an early detection scenario, where solid tumour biopsies are not available, we used matched tumour samples for validation purposes only, and never for guiding the search for copy number aberrations. We focused our analysis on whole chromosome arms to confidently distinguish positive signals from any remaining background noise (compare Fig. 2Aii, iii). Specifically, the denoised coverage signal in each arm was aggregated in each cancer plasma sample, as well as in each of nine age-matched non-cancer controls (CBS group; see 'Methods'). In the controls, the signal ranged from −0.34 to 0.22, while in the cancer samples, the range of coverage values was twice as wide (−0.71 to 0.42). To determine whether the aggregate coverage in each arm in a particular patient deviated significantly from the corresponding measurement in the controls (upward, indicating chromosomal gain, or downward, indicating chromosomal loss), we calculated a z-score for each arm as the number of standard deviations from the mean aggregate coverage of the same arm across the non-cancer CBS controls (see 'Methods'). In these controls, absolute z-scores were lower than 2.35, while in the cancer plasma samples absolute z-scores took values as high as 34.2 (Fig. 2Bi). Given these scores, p-values were calculated for each arm using an appropriate test statistic and a t-distribution with 8 degrees of freedom (9 controls minus 1), and they were corrected for multiplicity across all arms in each sample using the Benjamini-Hochberg procedure (see 'Methods'). Corrected p-values less than 0.05 (FDR < 5%) indicated significant loss or gain of chromosomal material in the corresponding arm in comparison to the CBS controls. Chromosomal arm alterations were detected in samples from all six cancer types in our cohort, with the number of altered arms per aberrant sample ranging from 1 to 29 (mean = 8.9, std. dev. = 8.1; Fig. 2Bi). Subsequently, we devised a sample-specific copy number aberration score by integrating the p-values across all aberrant arms in each sample using Stouffer's method. Overall, we detected aberrant arms in 15/36 colorectal, 3/8 oesophageal, 5/6 pancreatic, 3/5 renal, 2/4 ovarian and 1/2 breast cancer plasma samples resulting in 47.5% sensitivity (Fig. 2Bii). The integrated scores in the aberrant samples ranged from 3.0 to 18.1 (mean = 7.6, std. dev. = 4.1). The integrated CNA scores in the CBS control samples were all zero (100% specificity). Aberrant samples covered both early (Stage 1 or 2) and more advanced (Stage 3

or 4) cancer stages (Fig. 2Ci) and the median integrated scores increased monotonically with cancer stage (Fig. 2Cii).

To evaluate the capacity of the above approach to discriminate between non-cancer and cancer plasma samples, we conducted Receiver Operating Characteristic (ROC) analysis on synthetic data generated using actual clinical plasma samples as templates. We examined simulated ctDNA fractions ranging from 0.1% to 2% and for each ctDNA fraction, we simulated 1000 non-cancer controls and 1000 cancer plasma samples by admixing data from actual non-cancer CBS samples and a colorectal cancer plasma sample with 9% tumour burden (see Methods). For each sample in the synthetic case-control dataset, we calculated sample-specific integrated CNA scores, which were used in the construction of an ROC curve. The area under the ROC curve (AUC) at each simulated ctDNA fraction was used to assess classifier performance. The integrated CNA scores correlated with increasing ctDNA fractions 0.6% or higher, while 80% AUC was achieved at ctDNA fractions as low as 0.7% (Fig. 2D).

## Analysis of somatic mutation burden for ctDNA detection

Elucidating the profile of somatic mutations present in the plasma cfDNA has been a major research focus in the clinical study of ctDNA as an emerging biomarker for the detection of cancer and monitoring of disease progression. Towards this aim, the major obstacle has been the need to discriminate tumour-originating SNVs and indels from the much more abundant germline variants and sequencing errors. In order to overcome this problem, one group of methods focuses on deep targeted sequencing of cancer-type-specific panels and driver genes, combined with error-suppression methodologies[35]. Although potentially extremely sensitive, targeted approaches are constrained by the fact that they only sample a small number of human genome equivalents, possibly leading to an inflated false negative rate. In response, an alternative group of approaches centred around shallow WGS and compendiums of patient-specific somatic mutations to guide the analysis has been proposed, thus replacing depth with breadth of sequencing at the cost of increased sequencing noise[21].

To overcome these limitations, we adopted a deep (at least 80x) WGS approach for sensitive mutation detection without requiring matched biopsy samples to guide the analysis. Each plasma sample was a) paired with a matched germline sample for efficient removal of germline mutations, b) processed with bespoke software, which recognises changes induced by the TAPS process (mC>T) and differentiates these from C>T variants, c) cleaned up using a thorough filtering pipeline to remove sequencing artefacts, and d) further denoised by removing all variants shared with any of 21 age-matched non-cancer plasma samples from SCAN patients referred for cancer investigation with non-specific symptoms. All remaining non-synonymous variants were retained for further downstream analysis (see 'Methods' and Supplementary Fig. 2).

Each non-cancer control CBS plasma sample carried between 125 and 192 somatic mutations (mean = 151.1, std. dev. = 25.3; Fig. 3Ai). As expected, cancer plasma samples harboured on average a higher number of mutations, and showed higher variability (mean = 425.8, std. dev. = 747.3). Plasma samples from pancreatic cancer patients harboured the highest number of mutations on average (mean = 1771.2, std. dev. = 1770.2), followed by oesophageal (mean = 731.8, std. dev. = 1025.9), breast (mean = 385.5, std. dev. = 268.0), colorectal (mean = 288.0, std. dev. = 417.4) and renal (mean = 252.2, std. dev. = 224.0) cancer samples. Across all cancer plasma samples, 11730 genes harboured at least one somatic mutation. Among these, 8902, 1793 and 341 genes carried 1, 2 or 3 somatic mutations, respectively (Fig. 3Aii). As expected from whole genome data, among all mutated genes, only a small fraction of 493 (4.4%) genes were previously reported in COSMIC. Most mutations were non-COSMIC missense mutations, followed by nonsense and COSMIC missense mutations, and splice site mutations (Fig. 3Aiii and Supplementary Fig. 3). The vast

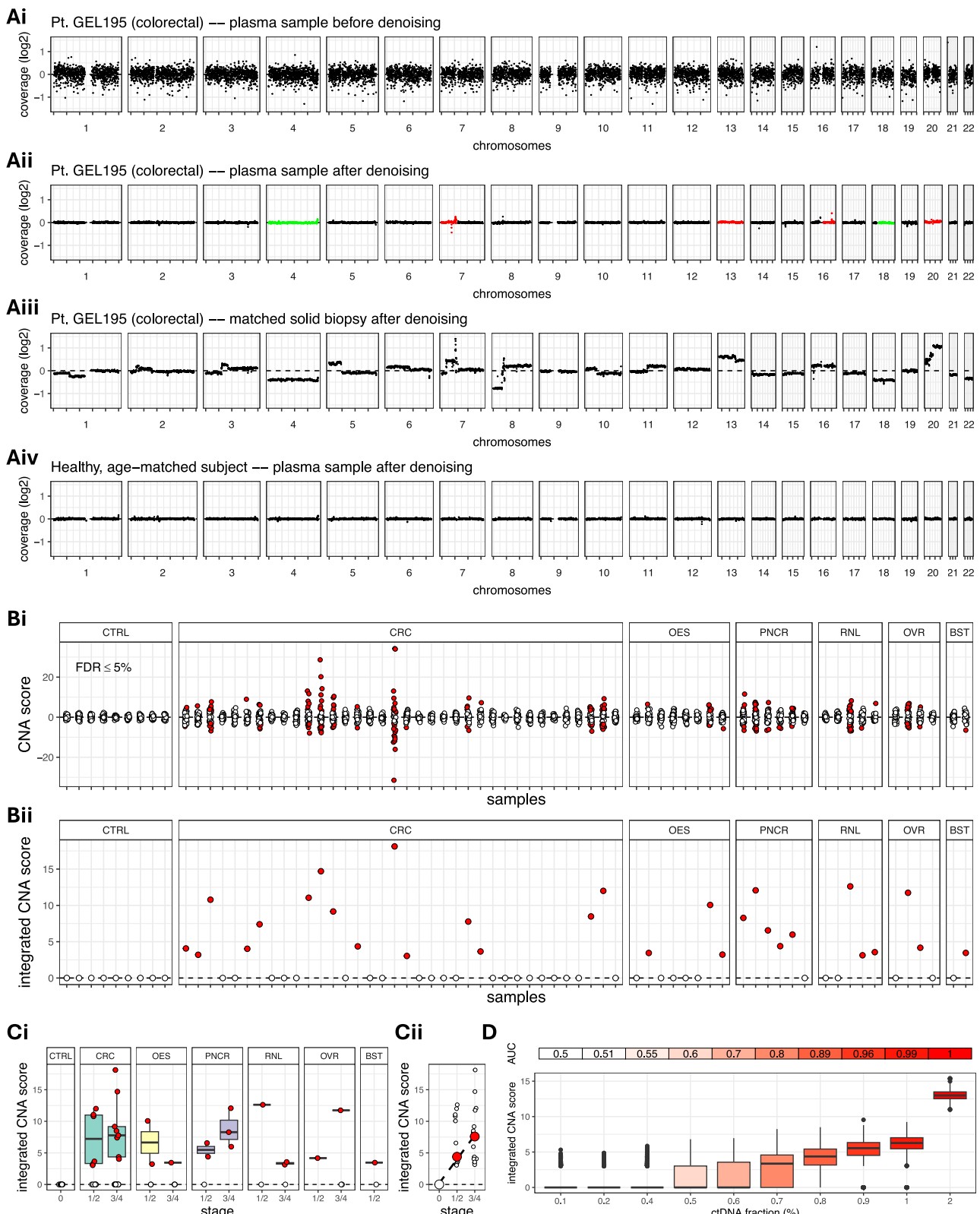

majority of these are likely representing acquired passenger events (which are expected to accumulate more rapidly in cancer samples than controls due to the higher division rate of tumour cells) that we used for cancer detection, as described below.

For each plasma sample, we compared the somatic mutation burden in each chromosome arm against the corresponding arms of the nine age-matched non-cancer CBS controls. Since the density of somatic

mutations is not uniform across the whole genome, a comparative analysis based on chromosome arms is expected to be more sensitive, compared to an analysis based merely on the total somatic mutation burden of the sample. For each chromosome arm in each sample, we calculated the $\log_{10}$ of the number of somatic mutations, we derived a z-score as the number of standard deviations from the mean of the same quantity across the CBS controls, and we calculated a p-value, which we

**Fig. 2 | Analysis of copy number aberrations (CNA). A** Coverage signal for patient GEL195 (colorectal cancer) before denoising (**Ai**), after denoising (**Aii**) and in the biopsy after denoising (**Aiii**). A non-cancer control is also shown for comparison (**Aiv**). The aggregated coverage signal in each chromosome arm in the plasma sample is compared against the corresponding arm in a cohort of non-cancer control plasma CBS samples in search of gains (red) or losses (green). Gains in chromosomes 7, 13, 16, 20 and losses in chromosomes 4 and 18 in the plasma sample of patient GEL195 reflect aberrations in the same chromosomes in the matched biopsy, although this has not been used for guiding the analysis. **Bi** Scores quantifying coverage imbalances in the chromosome arms of each cancer plasma sample compared to the non-cancer plasma CBS controls. In each sample, each circle corresponds to a different chromosome arm. Red circles indicate a gain or loss of chromosomal material. **Bii** Integrated CNA scores over all chromosome arms in each plasma sample. Red circles indicate the gain or loss of chromosomal material in the corresponding samples. 29 out of 61 cancer samples were correctly identified (sensitivity 47.5%). **C** Integrated CNA score against cancer stage and type (**Ci**) and monotonic increase of median integrated CNA score with cancer stage (**Cii**). **D** In silico assessment of CNA analysis performance at increasing ctDNA fractions. At each ctDNA fraction, we simulated 1000 non-cancer and 1000 cancer plasma samples using actual non-cancer and cancer plasma samples as templates (see 'Methods'). The area under the receiver operating characteristic (ROC) curve (AUC) was 80% at ctDNA fraction 0.7%. CTRL CBS controls ($n = 9$ subjects), *CRC* colorectal ($n = 36$ subjects), OES oesophageal ($n = 8$ subjects), PNCR pancreatic ($n = 6$ subjects), RNL renal ($n = 5$ subjects), OVR ovarian ($n = 4$ subjects), BST breast ($n = 2$ subjects). For each boxplot in (**Ci, D**) the box bounds, and centre correspond to the 25th, 50th (median), and 75th percentiles of the data in each corresponding group, and the whiskers extend to 1.5 times the interquartile range (IQR) above and below the box bounds. Source data is provided as a source data file.

corrected for multiplicity across all chromosome arms in each sample using the Benjamini-Hochberg method (see 'Methods'). A corrected *p*-value lower than 5% indicates a statistically significant difference in the mutation burden of the chromosome arm in comparison to the non-cancer CBS controls. A significantly increased chromosome arm mutation burden was detected in samples from all cancer types in our cohort, with the number of mutated arms per aberrant sample ranging from 1 to 39 (mean = 13.6, std. dev. = 15.9; Fig. 3Bi). Subsequently, we devised a sample-specific somatic mutation score by integrating the *p*-values across all significantly mutated arms in each sample using Stouffer's method. Overall, we detected arms with increased mutation burden (*p* value < 5%) in 17/36 colorectal, 5/8 oesophageal, 5/6 pancreatic, 2/5 renal, 1/4 ovarian and 2/2 breast cancer plasma samples resulting in 52.5% sensitivity (Fig. 3Bii). The integrated scores in the aberrant samples ranged from 2.63 to 23.2 (mean = 10.4, std. dev. = 7.2) and were zero in all non-cancer CBS controls (100% specificity). Aberrant samples covered both early (Stage 1 or 2) and more advanced (Stage 3 or 4) cancer stages (Fig. 3Ci), while median integrated scores were moderately correlated with cancer stage (Spearman's *r* = 50%; Fig. 3Cii). Further evaluation of the discriminatory performance of the above analytical approach using ROC analysis on synthetic data (see 'Methods') indicates AUC at least 74% at ctDNA fractions 1% or higher (Fig. 3D). This is consistent with a depth of coverage at 100×, which implies a minimum variant allele fraction (VAF) for any mutated locus of 1% (at least one mutated read in a locus covered by 100 reads).

### Analysis of methylation signals for ctDNA detection

DNA methylation is an epigenetic mechanism that can regulate gene expression. When located in a promoter, it typically acts to supress gene expression. Thus, hypomethylation and hypermethylation can lead to increased expression of oncogenes or decreased expression of tumour suppressor genes, respectively. Abnormal DNA methylation patterns are associated with all aspects of cancer pathophysiology, from tumour initiation to tumour progression and metastasis, making DNA methylation abnormalities one of the hallmarks of cancer that can be exploited for disease diagnosis, treatment and monitoring[36–41].

Current approaches for investigating the methylome using plasma cfDNA[3,39–41] rely on tissue-specific methylation signatures, which provide a reference set against which the methylation profile of any case of interest is compared to. In this study, we extracted data from several TCGA studies to identify a set of hypermethylated regions, each containing at least three differentially methylated CPGs (Supplementary Data 2). We chose to focus on hypermethylation markers, where baseline methylation should be close to zero in non-cancer controls and only successful conversion by the TAPS chemistry would result in a positive signal. Subsequently, we identified fragments in each cfDNA sample overlapping at least three CPGs in any of these regions and we calculated the overall methylation level of the fragment. We decided to follow a fragment- rather than a locus-centric approach, since it is known that in low ctDNA fraction settings, this increases sensitivity[40]. Fragments with higher than 80% methylation were classified as tumour-originating and the total fraction of tumour-originating fragments was calculated in each region. An additional level of denoising was added by removing all regions containing at least one tumour-originating fragment in any of the 21 non-cancer SCAN controls (Supplementary Data 2).

Next, we compared the fraction of tumour-originating fragments in each region between each plasma sample and the 9 age-matched non-cancer CBS controls. For each region in each sample, we calculated the logit of the fragment fraction, we derived a z-score and a corresponding *p*-value, which was corrected for multiplicity across all regions in each sample using the Benjamini-Hochberg method (see 'Methods'). The number of regions with significantly higher burden of tumour-originating fragments per sample ranged from 1 to 371 (mean = 113.6, std. dev. = 134.1; Fig. 4Ai) and, from the *p*-values of these regions, an integrated methylation score was derived for each sample using Stouffer's method. Overall, we detected regions with significantly higher methylation burden (*p*-value < 5%) in 18/36 colorectal, 2/8 oesophageal, 1/6 pancreatic, 4/5 renal, 3/4 ovarian and 0/2 breast cancer plasma samples resulting in 45.9% sensitivity (Fig. 4Aii). The integrated methylation scores in the aberrant samples ranged from 3.8 to 64.0 (mean = 24.5, std. dev. = 17.7) and were zero in all CBS controls (100% specificity). Aberrant samples covered both early (Stage 1 or 2) and more advanced (Stage 3 or 4) cancer stages (Fig. 4Bi) and the median integrated scores increased monotonically with cancer stage (Fig. 4Bii). ROC analysis on synthetic data (see 'Methods') indicated an AUC value of 87% at a ctDNA fraction of 0.9% (Fig. 4C).

### Integration of multiple genomic modalities for ctDNA detection

The genomic data modalities analysed above provide three independent and complementary assessments of tumour content in the plasma. We reasoned that combining these data modalities may enrich the available signal and increase the sensitivity of detection of ctDNA in each plasma sample, particularly in cases where not all three types of abnormalities are detectable due to the sparsity of ctDNA. In each sample, given the sample-specific *p*-values for each data modality, an integrated multimodal score and corresponding *p*-value can be readily calculated using Stouffer's method. Although in this study all three modalities are combined using equal weights, unequal weights specific to each genomic data type can also be introduced, if necessary. Furthermore, this approach is applicable even if not all three modalities are available in a particular sample. In our cohort, multimodal scores across aberrant plasma samples ranged from 1.8 to 36.5 (mean = 11.8, std. dev. = 9.8) and they were all zero across the 9 age-matched non-cancer CBS controls (specificity 100%). Significant *p*-values (<5%) indicating the presence of ctDNA were calculated in 29/36 colorectal, 7/8 oesophageal, 5/6 pancreatic, 5/5 renal, 4/4 ovarian and 2/2 breast cancer plasma samples resulting in a substantially increased sensitivity of 85.2%, compared to the sensitivity of each independent data modality (Fig. 5A). Positive results covered both early- (stages 1 or 2) and

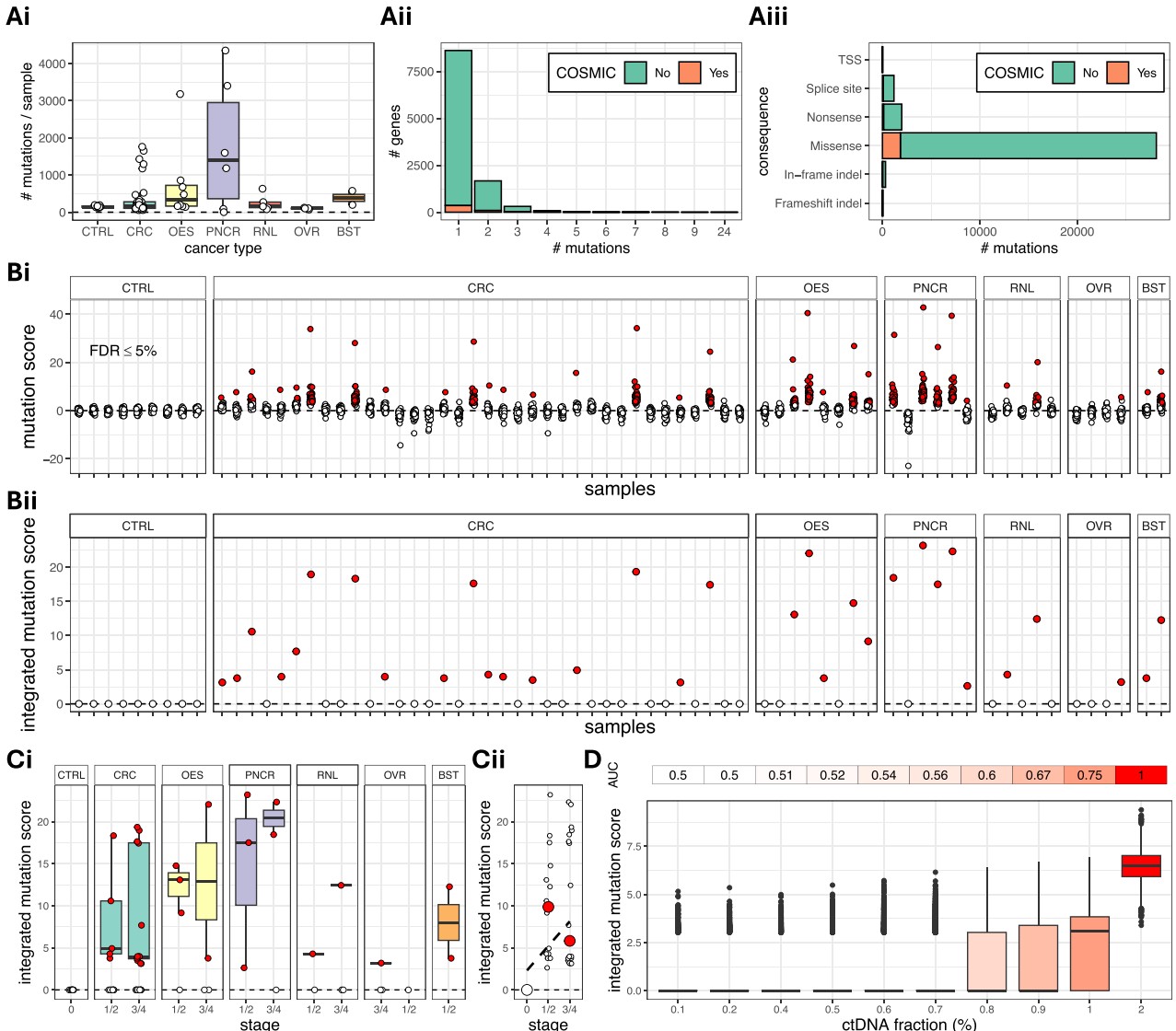

**Fig. 3 | Burden analysis of somatic single nucleotide variants (SNVs) and INDELs. A** Somatic mutation burden in different cancer types and in non-cancer CBS controls (**Ai**), distribution of mutation numbers across genes (**Aii**) and consequences of mutations (**Aiii**). In (**Ai**), each circle corresponds to a different plasma sample. **Bi** Scores quantifying mutation burden imbalances in the chromosome arms of each cancer plasma sample compared to the non-cancer plasma CBS controls. In each sample, each circle corresponds to a different chromosome arm. Red circles indicate a difference in the somatic mutations burden of the chromosome arm in relation to the same arm in the CBS controls. **Bii** Integrated somatic mutation scores over all chromosome arms in each plasma sample. A red circle indicates a higher mutation burden in the corresponding sample, when compared to the CBS controls. We identified correctly 32 out of 61 cancer plasma samples (sensitivity 52.5%). **C** Integrated somatic mutation scores against cancer stage and

type (**Ci**) and moderate correlation between median integrated somatic mutation score and stage (**Cii**); Spearman's $r = 50\%$). **D** In silico validation of somatic mutation analysis at increasing ctDNA fractions. At each ctDNA fraction, we simulated 1000 controls and 1000 cancer plasma samples using actual non-cancer and cancer plasma samples as templates (see 'Methods'). The area under the receiver operating characteristic (ROC) curve (AUC) is 74% at ctDNA fraction 1%. CTRL CBS controls ($n = 9$ subjects); CRC colorectal ($n = 36$ subjects), OES oesophageal ($n = 8$ subjects), PNCR pancreatic ($n = 6$ subjects), RNL renal ($n = 5$ subjects), OVR ovarian ($n = 4$ subjects), BST breast ($n = 2$ subjects). For each boxplot in (**Ai, Ci, D**), the box bounds, and centre correspond to the 25th, 50th (median) and 75th percentiles of the data in each corresponding group, and the whiskers extend to 1.5 times the interquartile range (IQR) above and below the box bounds. Source data is provided as a source data file.

late-stage (stages 3 and 4) cancer cases (Fig. 5Bi). Median multimodal scores increased monotonically with cancer stage (Fig. 5Bii), while ROC analysis using synthetic data indicates an AUC of 86% at ctDNA fractions as low as 0.7% (Fig. 5C).

Further validation using leave-one-out cross-validation (LOO-CV; see 'Methods') indicates 85.2% sensitivity and 88.8% specificity (implying a balanced accuracy of 87%), and an AUC of 83.5%, when all three data modalities are considered (Supplementary Fig. 4). In addition, we used LOO-CV to assess four additional methodologies for integrating different data modalities for cancer status prediction; specifically, Fisher's method (a popular alternative to Stouffer's),

Logistic Regression, Random Forest and Support Vector Machine, all of which are commonly used in the statistics and machine-learning community (see 'Methods'). The best performing methods were Stouffer's and Fisher's (AUC 83.5%), followed by Logistic Regression (AUC 81.3%), Random Forest (80.7%) and Support Vector Machine (AUC 66.3%; Supplementary Fig. 5).

Finally, we developed a multi-class classifier by integrating all three data modalities (copy number aberrations, somatic single nucleotide variants and indels, and methylation signals) for predicting cancer type (see 'Methods'), and we validated its performance using LOO-CV (Supplementary Fig. 6). The multi-class classifier is a Support

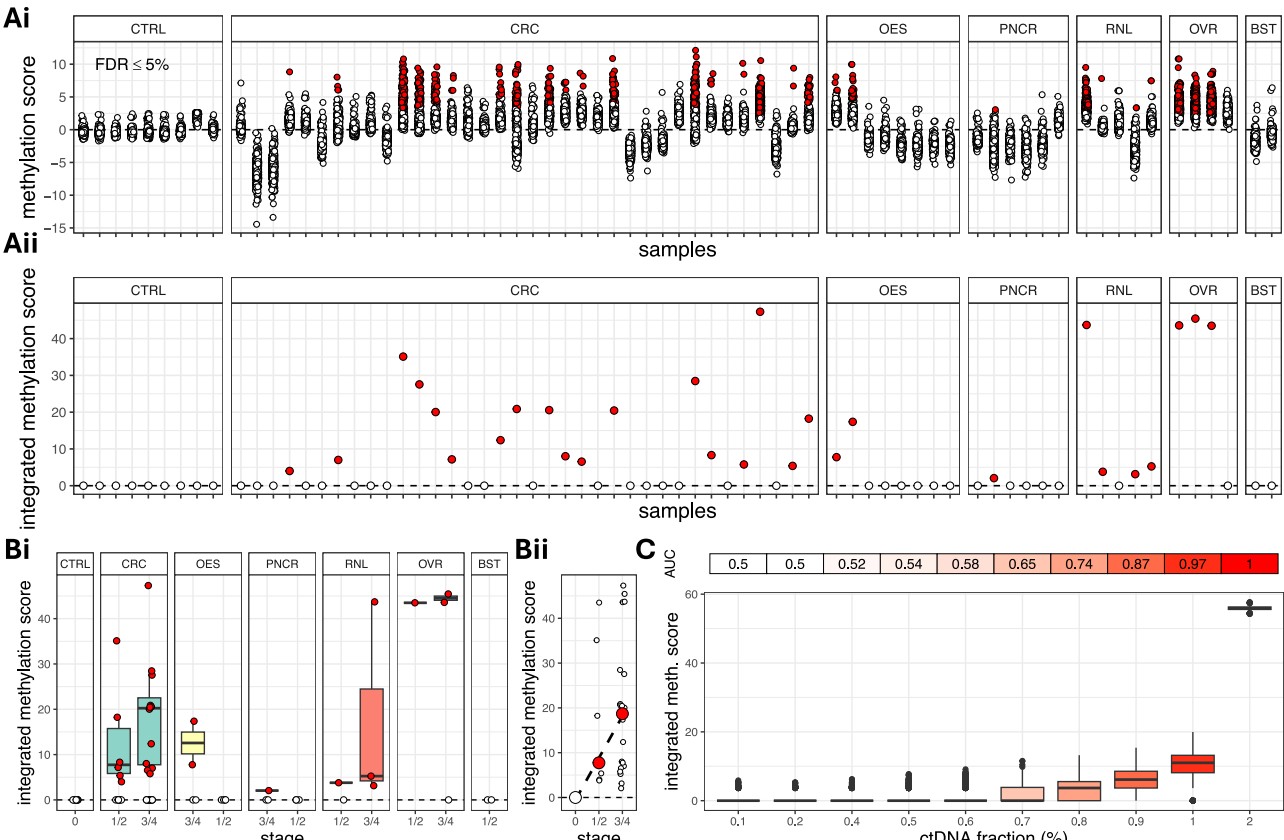

**Fig. 4 | Overview of methylation analysis. Ai** Scores quantifying imbalances in the methylation burden in any of 377 regions (extracted from TCGA; see 'Methods') in each cancer plasma sample compared to the non-cancer plasma CBS controls. Each circle corresponds to a different region and red circles indicate over-methylation of the corresponding regions between the cancer plasma and the CBS controls. **Aii** Integrated methylation scores over all regions in each plasma sample. A red circle indicates an over-methylated plasma sample, when compared to the CBS controls. We identified correctly 28 out of 61 cancer plasma samples, which corresponds to a 45.9% sensitivity. **B** Integrated methylation scores against cancer stage and type (**Bi**) and monotonic increase of median integrated methylation scores with cancer stage (**Bii**). **C** In silico validation of methylation analysis at increasing ctDNA fractions. At each ctDNA fraction, we simulated 1000 non-cancer and 1000 cancer plasma samples using actual non-cancer and cancer plasma samples as templates (see 'Methods'). The area under the receiver operating characteristic (ROC) curve (AUC) is 87% at ctDNA fractions 0.9%. CTRL CBS controls ($n = 9$ subjects); CRC colorectal ($n = 36$ subjects), OES oesophageal ($n = 8$ subjects), PNCR pancreatic ($n = 6$ subjects), RNL renal ($n = 5$ subjects), OVR ovarian ($n = 4$ subjects), BST breast ($n = 2$ subjects). For each boxplot in (**Bi, C**), the box bounds, and centre correspond to the 25th, 50th (median) and 75th percentiles of the data in each corresponding group, and the whiskers extend to 1.5 times the interquartile range (IQR) above and below the box bounds. Source data is provided as a source data file.

Vector Machine, and it correctly identifies the cancer origin in 25/36 colorectal, 4/8 oesophageal, 3/4 ovarian, 2/6 pancreatic and 0/5 renal cancer cases, leading to an overall balanced classification accuracy of 71.7%. At the same time, we correctly identified as negative 8/9 non-cancer controls and as positive 56/59 cancer cases resulting in 94.9% sensitivity and 88.8% specificity. It is interesting to observe that among the 377 hypermethylated TCGA regions used in our analysis, only 4 were specific to renal cancer (see Methods), which may explain our inability to correctly classify any of the 5 corresponding cases in our cohort (although they are correctly identified as cancer cases). Notice that we did not consider two cases with breast cancer, because it is not possible to conduct cross validation of the multi-cancer classifier with only two cases in any of the cancer groups.

**Multimodal ctDNA detection for post-operative MRD and adjuvant therapy response tracking in colorectal cancer without matched tumour**

Up to this point, we focused our analysis on treatment-naive and non-cancer plasma samples, aiming to imitate an early detection scenario. In order to evaluate our approach for tracking disease progression[42–45], we further analysed pre- and post-operative plasma samples from 10 colorectal cancer patients with detectable ctDNA in the pre-operative sample, half of which had also received adjuvant therapy following surgical resection. For each plasma sample in each patient, integrated ctDNA scores and p-values from all three data modalities were calculated as for the pre-operative samples. A p-value threshold of 5% was taken to indicate the presence of ctDNA in the plasma. As before, matched tumour biopsies, if available, were not used for guiding the analysis, since this carries the risk of missing recurrence due to clonal evolution of the primary tumour or the presence of a second primary. Besides, it has been shown that in the real-world, there are significant delays in accessing tumour tissue biopsies for monitoring.

Five patients did not receive adjuvant therapy. In one of them, (GEL193; Fig. 6A), ctDNA was detectable in the plasma 1 year after surgery. The patient was diagnosed with inoperable metastatic rectal cancer three years later and a possible lung adenocarcinoma on radiological examination. Two patients (GEL066, Supplementary Fig. 7Ai; GEL339, Supplementary Fig. 7Aii) had detectable ctDNA in the plasma 2 years and 6 months after surgery, respectively. They were found to have tubular adenomas with low-grade dysplasia on routine follow-up 2 years and 7 months after the last post-surgery blood samples were collected, respectively. The remaining two patients permanently cleared ctDNA 3 months and 16 months after surgery, respectively (Supplementary Fig. 7Aiii, iv). One of them (GEL107) is alive 8 years after the last blood sample was taken, while the other (GEL197) died from sudden cardiac death after 5 years.

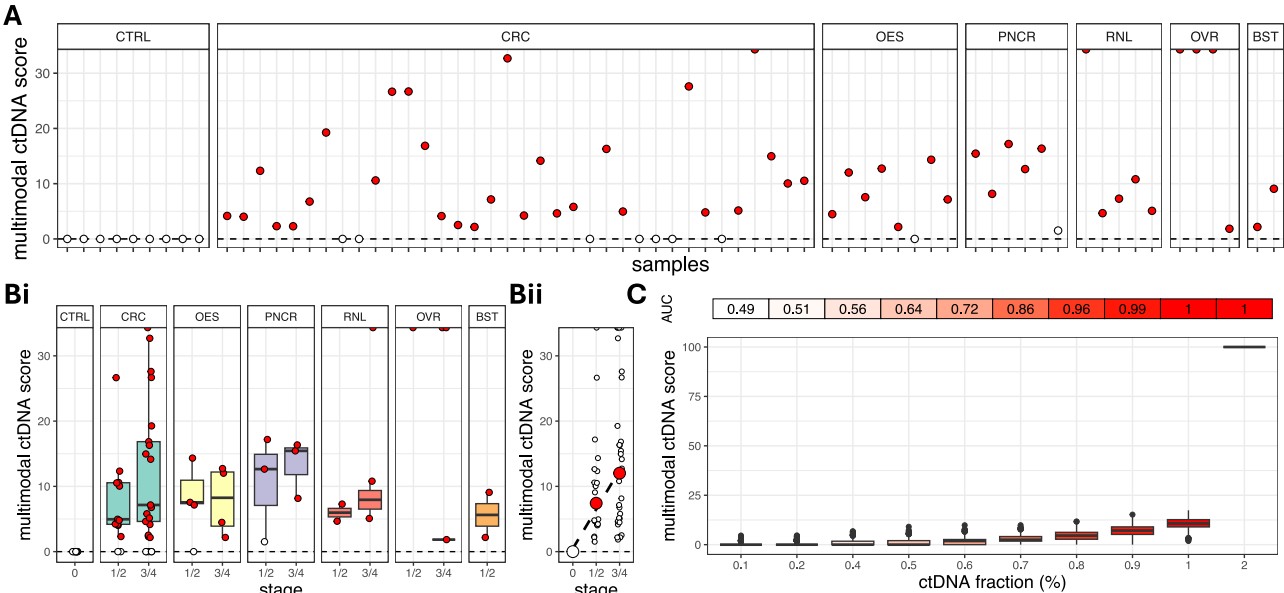

**Fig. 5 | Integration of genomic data modalities for ctDNA detection.**
**A** Multimodal scores for the quantification of plasma ctDNA generated from the integration of copy number aberrations, somatic SNVs and INDELs, and methylation signals in each plasma sample. A red circle indicates a higher ctDNA burden in comparison to the non-cancer CBS controls. 52 out of 61 cancer plasma samples were correctly identified as such, which corresponds to 85.2% sensitivity. This is higher than the sensitivity of any of the three data modalities. **B** Multimodal scores against cancer stage and type (**Bi**) and monotonic increase of median multimodal scores with cancer stage (**Bii**). **C** In silico validation of multimodal analysis at increasing ctDNA fractions. At each ctDNA fraction, we simulated 1000 controls

and 1000 cancer plasma samples using actual non-cancer and cancer plasma samples as templates (see Methods). The area under the receiver operating characteristic (ROC) curve (AUC) is 86% at ctDNA fractions 0.7%. CTRL CBS controls ($n = 9$ subjects), CRC colorectal ($n = 36$ subjects), OES oesophageal ($n = 8$ subjects), PNCR pancreatic ($n = 6$ subjects), RNL renal ($n = 5$ subjects), OVR ovarian ($n = 4$ subjects), BST breast ($n = 2$ subjects). For each boxplot in (**Bi**, **C**), the box bounds, and centre correspond to the 25th, 50th (median) and 75th percentiles of the data in each corresponding group, and the whiskers extend to 1.5 times the interquartile range (IQR) above and below the box bounds. Source data is provided as a source data file.

Five patients received adjuvant therapy after surgery. GEL282 did not have detectable ctDNA immediately after the last cycle of adjuvant treatment (Fig. 6Bi). However, a low positive ctDNA burden was detected 5 months later, which correlated with the presence of tubular adenomas with low grade dysplasia in the sigmoid colon at around the same time. Cases GEL432 (Fig. 6Bii) and GEL205 (Supplementary Fig. 7Bi) did not have detectable ctDNA after the end of treatment and they were both alive 6 years and 4 years after the last plasma sample was collected, respectively.

For case GEL195 (Supplementary Fig. 7Bii), a post-operative blood sample collected 3 months after surgery indicated the presence of ctDNA in the plasma, although there was a 74% reduction in tumour burden compared to the pre-surgery sample (multimodal ctDNA score before surgery: 16.86, after surgery: 4.34). Since the sample was taken 1½ months after the first cycle and ~4 months before the last cycle of adjuvant treatment, we presume that there would be no detectable ctDNA in the plasma after the end of therapy. The patient was still alive ~8 years after the last blood sample was collected. Finally, case GEL223 (Supplementary Fig. 7Biii) did not show detectable ctDNA levels 8 months after the last cycle of adjuvant therapy, despite the diagnosis of prostate adenocarcinoma 1 month earlier. This was treated with hormone therapy and radiotherapy, and the patient was still alive 6½ years after the last plasma sample was collected.

Overall, presence or absence of ctDNA in the last plasma sample was correlated with adjuvant therapy or clinical outcome in 9 out of 10 cases (the exception was case GEL223). Furthermore, in the cases for which we had plasma samples after surgery or after the end of adjuvant therapy ($n = 9$ cases), event-free survival (i.e., no recurrence, metastasis, or pre-cancerous adenomas) was correlated with the absence of detectable ctDNA post-treatment (HR 8.2; 95% CI 1.3–53.1; two-sided log-rank test $p$ value = 0.02; Fig. 6Ci, ii). Follow-up samples from more patients would be necessary to increase our confidence in this

interesting result and to estimate with higher precision the association of post-treatment residual ctDNA with event-free survival.

## Discussion

After sequencing TAPS-treated ctDNA at a target coverage depth of at least 80x, we integrated genome-wide information from CNAs, acquired point mutations and indels, and methylation changes with a strict de-noising strategy to reveal cancer signals with high sensitivity and specificity. We sequenced plasma and matched germline samples from 61 cancer and 30 non-cancer subjects, as well as several follow-up plasma samples from 10 of the colorectal cancer patients for a total of 214 samples. We hope that this large, deeply sequenced dataset will be a valuable resource to the cfDNA research community. Furthermore, we show that comprehensive multimodal whole-genome TAPS of ctDNA including CNV, SNV and DNA methylation analysis at 80x or higher can improve the detection of cancer signals from liquid biopsies.

We demonstrate that we can successfully call acquired SNVs from TAPS-treated samples in a single sequencing run using bioinformatics to correct for C>T changes. As we only had paired tissue samples available for a minority of cases, future studies are needed to understand whether the mutation spectrum seen in plasma using this approach is representative of that seen in corresponding tissue biopsies.

While future validation in larger data sets of different cancer types and stages is needed, we provide proof-of-principle that deep whole genome sequencing combining depth with breadth of sequencing to allow integration of information from various modalities (copy number aberrations, single nucleotide variants, insertions/deletions, and methylation signals) is sensitive.

Circulating tumour DNA is highly fragmented, of low abundance and highly diluted in circulating germline DNA originating from

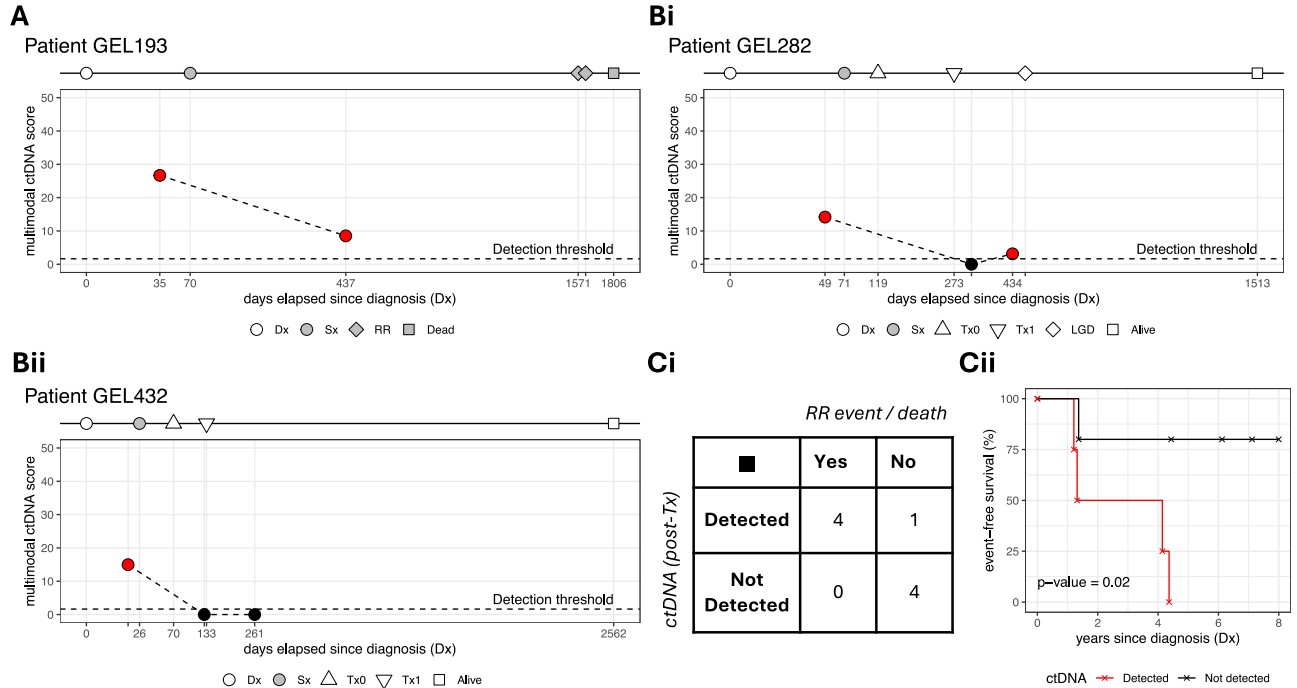

**Fig. 6 | Multimodal ctDNA detection for post-operative MRD and adjuvant therapy response tracking in colorectal cancer without matched tumour.**
**A** Tracking post-operative MRD in case GEL193. ctDNA was detectable in the plasma 1 year after surgery and this correlated with inoperable metastatic rectal cancer and a possible lung adenocarcinoma suggested through radiological examination, both of which were recorded ~3 years after the post-operative plasma sample was collected. **B** Tracking response to adjuvant therapy following surgery. Case GEL282 (**Bi**) did not have detectable ctDNA immediately after the last cycle of treatment. However, low ctDNA burden was detected 5 months later, which correlated with the presence of tubular adenomas with low grade dysplasia in the sigmoid colon at around the same time. Case GEL432 (**Bii**) did not have detectable ctDNA shortly

after the last cycle of treatment and was still alive ~6 years after the last plasma sample was collected. **C** Confusion matrix (**Ci**) and event-free (i.e., no recurrence, metastasis, or precancerous adenomas present) survival (**Cii**) in 9 patients with colorectal cancer. In 8 out 9 patients, ctDNA burden after the end of surgery and/or adjuvant therapy correlated with the presence/absence of clinical events, such as recurrence or pre-cancerous adenomas (**Ci**). Absence of ctDNA detection after the end of surgery/adjuvant treatment correlated (hazard ratio: 8.2; 95% CI: 1.3–53.1; two-sided log-rank test *p* value = 0.02) with longer survival times (**Cii**). Dx diagnosis, Sx surgery, Tx0 first cycle of adjuvant therapy, Tx1 last cycle of adjuvant therapy, RR recurrence, LGD low-grade dysplasia. Source data is provided as a source data file.

peripheral blood mononuclear cells. In addition, some cancers might not have many copy number changes. Instead, they display a characteristic acquired SNV or methylation signature profile. Others (~25% of cancers worldwide) are infection-related and shred viral-human DNA hybrid fragments into plasma. A main advantage of our approach is therefore that it considers cancer heterogeneity by using different types of aberrant signals that are enriched for somatic (acquired) events to refine the probability for a cancer diagnosis.

This case-control design aimed to explore the potential utility of our approach for triaging patients presenting with symptoms of cancer. We used samples from patients who were referred to surgery with curative intent to enrich for earlier stage cancers, as well as samples from patients presenting with non-specific symptoms of cancer, who remained without a cancer diagnosis for at least two years following the blood draw, and who were used as controls for signal denoising. If we were to apply the cancer prevalence of 6.7% observed in SYMPLIFY to model our test performance and assume a test sensitivity and specificity of 85.2% and 88.8%, respectively (which are the values we observed in this study, based on 9 negative CBS controls and 61 cancer cases), then the negative predictive value (NPV) would be 98.8% in this group of patients. A negative test result would reduce the risk of cancer down from 6.7% to 1.2%. If a cohort with a lower pre-test probability was investigated in primary care pre-referral, such as a cohort with a pre-test probability of 3% (the referral threshold used in the NHS), the NPV would be 99.5% and the post-test probability following a negative test would decrease from 3% down to 0.5%. Considering these encouraging predictions, prospective studies in primary and secondary care are required to confirm these estimates.

Therefore, the results of our study presented here pave the way to a future larger scale validation in people with high-risk features. These could be defined by age[16], non-specific symptoms[5] or genetic cancer predisposition, life-style habits, or the refusal to participate in any of the existing more invasive screening procedures for cervical, breast, lung or colorectal cancer. Past and ongoing studies led by others have already highlighted some of the practical challenges associated with early cancer detection from liquid biopsies, particularly the potential psychological, emotional and financial "toxicity" from false positive results[2,46]. For MCED tests to enter clinical practice, future efforts are needed to further improve test attributes, especially positive predictive values. Importantly, long-term follow-up of the cohort study Detect-A already suggests that overall survival of patients correctly identified with cancer from liquid biopsy screening is improved while emotional or physical morbidity from false positive results recovers soon after testing[47]. Additional and prospective data will be hopefully available from longer follow-up of the GRAIL-NHS study.

The results presented here must be validated in randomised controlled studies of people at risk before implementation into public healthcare systems that rely on re-imbursement from national or private insurance contributions. Once, randomisation demonstrates clinical utility and cost-effectiveness, a complementary and important step in the validation process is the post-approval real-world (RW) data collection (Phase 4). Although data is often of lesser quality and incomplete compared to formal prospective studies, RW data will reveal important implementation challenges, for example: who the test is prescribed for (e.g., at risk individuals and what type of at-risk

individuals? How often is the test prescribed? How are false positive results handled in the RW?).

It remains to be seen whether additional refinements of the bioinformatics pipeline integrating fragment size distribution, mutational signatures, telomere length and other genome-wide cancer signals increase sensitivity further. As we have generated whole (epi-) genome data, we investigated the capacity of our methodology to also predict cancer origin. This analysis is limited by the fact that our methylation signatures are based on array data (covering approximately only 1% of the human genome) for a small set of cancer types from TCGA. This data comes mainly from colorectal cases, as this is where the TCGA dataset was most complete. To perform more precise tissue-of-origin analysis across many different tumour types, we need to have access to a methylation atlas of normal tissue generated using whole-genome TAPS. So far, such methylation atlases (including the TCGA methylation array data) have only been generated using bisulphite sequencing. A TAPS-specific methylation atlas is in preparation (Chun-Xiao Song, personal communication) to enhance the accuracy of TAPS-derived tissue-of-origin (TOO) prediction. However, the debate whether TOO prediction is useful in clinical practice or whether it leads to an increase in false positive cancer calls is on-going.

Furthermore, deep sequencing (at 80x) of both plasma and matching germline samples could pose an obstacle in the wider adoption of this approach, particularly in resource-restricted clinical settings. For this reason, future research should also focus on determining the minimum feasible depth of coverage for this type of study using down-sampling experiments made possible by the deep whole-genome sequencing dataset we present here, as well as on the comparison against potentially cheaper long-read sequencing technologies.

In conclusion, we indicate that our approach to multi-modal ctDNA analysis using deep whole genome sequencing combined with TAPS detects cancer signals in early- and late-stage cancer with high accuracy. The next step will be to perform prospective studies in unselected consecutive cases to fully establish diagnostic performance.

## Methods

The research presented in this paper complies with all relevant ethical regulations (see 'Ethics section', below).

### Ethics

The National Health Service (NHS) Health Research Authority South Central−Oxford C Research Ethics Committee approved this study, and all research was performed in accordance with relevant regulations and guidelines and with the Declaration of Helsinki. Written informed consent was obtained for patients recruited into the 100,000 Genomics England (GEL) pilot study and the rapid diagnostic clinic research pathway in Oxford called Suspected CANcer (SCAN) according to Oxford Radcliffe Biobank (ORB) guidelines (Oxford C Research Ethics Committee Number: 19/SC/0173). For 9 additional non-cancer controls (CBS group), blood was received from Cambridge Bioscience Human Blood Products Supply Service, where comprehensive informed written consent was provided in accordance with UK ethics and consent regulations.

### Study design and clinical cohorts

We conducted a diagnostic accuracy study using a case-control design (Fig. 1 and Supplementray Data 1). Patients were eligible for recruitment if they were aged 18 years or above, willing, and able to give informed consent for participation in the study and were referred for urgent investigation for a possible gynaecological, lower GI or upper GI or renal cancer or to a rapid diagnostic centre (RDC) with non-specific symptoms that might be due to cancer. Referral criteria for each pathway were as summarised in the National Institute for Health and

Care Excellence (NICE) Guideline 12 Suspected cancer: recognition and referral (NG12) (Supplementary Material)[48]. Patients could not enter the study if they had a history of invasive or haematological malignancy diagnosed within the preceding 3 years or if they were taking cytotoxic or demethylating agents that might interfere with test performance. Only samples of patients ($n = 61$; GEL Cancer Pilot group) with confirmed cancer diagnosis who were advised to undergo surgery with curative intent were selected. For non-cancer controls, only samples of patients referred with non-specific symptoms of cancer who had not developed cancer within the subsequent two years were chosen for signal-to-noise control ($n = 21$; SCAN group). Asymptomatic age and sex-matched people served as negative controls ($n = 9$; CBS group).

For measurable disease monitoring, we chose a cohort of 10 patients with colorectal cancer undergoing surgery with or without adjuvant chemotherapy, which had detectable ctDNA at the pre-treatment plasma sample. Post-surgery samples were taken from all patients at least 6 weeks after surgery. For 5 patients, additional follow-up samples were obtained. In total, WGS data from ctDNA was available from 26 samples.

### Clinical sample collection

Blood samples were collected into $K_2$EDTA tubes in all cases for germline DNA (gDNA) extraction from normal peripheral blood leucocytes. For GEL pilot patients, blood was collected into additional $K_2$EDTA tubes and processed within 4 h for plasma extraction. For all other cases, blood was collected into PAXgene Blood ccfDNA tubes and processed within 72 h of venepuncture. Blood samples were centrifuged for 10 min at 1600 g to separate the plasma, which was subsequently aliquoted into fresh tubes and underwent a second centrifugation for 10 min at 4500 g. The plasma supernatants were stored in aliquots at −80 °C prior to extraction. Tissue samples were obtained during surgical resection and Fresh Frozen (FF) until required for extraction.

### DNA extraction

cfDNA was extracted using the QIAamp circulating nucleic acid kit (Qiagen), according to the manufacturer's instructions. Input plasma volumes ranged from 4 ml to 25 ml per sample, where the maximum input volume per extraction was 5 ml. Where more than one extraction was carried out per patient, extracted cfDNA was pooled. DNA was first eluted into 30 μl of buffer AVE and subsequently a second elution of 30 μl to maximise yield. gDNA was extracted using the QIAamp Blood Mini kit (Qiagen), according to the manufacturer's instructions. FF tissue samples initially underwent disruption and homogenization using a TissueRupter and DNA tumour DNA (tDNA) was extracted using the QIAamp Allprep Mini kit (Qiagen), according to the manufacturer's instructions. All samples were quantified using the Qubit Fluorimeter (ThermoFisher) high sensitivity dsDNA assay.

### Library preparation and sequencing

A methylated control (control A) and an unmethylated control (control B) were spiked-in to the extracted DNA. 1 μg of GL and FF DNA spiked with controls were fragmented to 450 bp using the M220 focused-ultrasonicator (Covaris). GL and FF DNA were size selected for 300−500 bp fragments using 0.55X followed by 0.75X ratios of sample to Agencourt AMPure XP Beads (Beckman Coulter). 50 ng of cfDNA were spiked with the same two controls previously fragmented to 150 bp. End repair and adapter ligation adjusted with a 1:10 dilution of adapters for cfDNA, were performed using the KAPA HyperPrep Kit (KAPA Biosystems), according to the manufacturer's recommendations. Post adapter ligation, an AMPure bead purification step was done using 0.8X ratio of beads to sample and DNA eluted in buffer EB. DNA oxidation used reagents and protocols supplied by Exact Sciences Innovation. Oxidation was followed by an incubation with Proteinase K

(0.8U) (New England Biolabs (NEB)) at 50 °C for 30 min, stopping the oxidation. This was followed by an AMPure bead clean-up using a 1.8X bead to sample ratio. Reduction was completed using borane compound and other reagents and protocols supplied by Exact sciences Innovation. Subsequently, libraries were purified using AMPure XP beads at a 1.8X bead to sample ratio. A first extension was completed with an enzyme mix from Exact Sciences Innovation before library amplification was performed using KAPA HiFi HotStart Uracil+ ReadyMix Kit (KAPA Biosystems), according to the manufacturer's instructions, with the exception that i5 and i7 NEBNext Multiplex Dual Oligos for Illumina (NEB) were used. GDNA and tDNA libraries underwent 4 PCR cycles, while 6 PCR cycles were performed on cfDNA libraries. Final libraries were purified using AMPure beads at a 1X ratio and DNA was eluted into buffer EB. To assess the conversion of 5mC, amplification of a 194 bp DNA fragment from the methylated lambda control was performed, using the primers F_5′-GCTGGGGAACTA-CAGGCT-3′ and R_5′-AGAACCAGAACTCAAACTGTAC-3′ (Integrated DNA Technologies). A PCR master mix was made containing 1 μl 10x Standard Taq Buffer, 0.5 μl 10 mM dNTPs, 0.5 μl 10 mM primers, 0.25 μl Taq polymerase, 2 ng DNA template and the final volume made up to 50 μl with nuclease-free water. Thirty-five cycles of PCR were carried out (1 cycle of 95 °C 30 s, 35 cycles of 95 °C, 48 °C 30 s, 68 °C 1 s, followed by 1 cycle of 68 °C 5 s). Once complete, a restriction-enzyme digest mix was set up containing 1 μl 10X Cutsmart buffer, 0.2 μl TaqI, 5 μl PCR product and made up to 10 μl with nuclease water. This mix was incubated at 65 °C for 30 min and the products analysed using agarose gel electrophoresis. Shallow sequencing runs for initial quality metrics were performed using a MiniSeq (Illumina). Paired-end (2 × 150bp) runs utilised the MiniSeq High Output Reagent Kit (300-cycles) aiming for a depth of at least 0.4x. Whole-genome sequencing was performed on a NovaSeq 6000 (Illumina) (paired-end reads–2 × 150) aiming at 30x sequencing depth for GL and at least 80x for FF and cfDNA on an S4 flowcell (Illumina) (Supplementary Fig. 8).

### Read Alignment
Paired-end sequencing reads were aligned to the reference genome GRCh38 using `bwa v0.17.7`, facilitated by `asTair v3.3.2`. The alignment process generated BAM files containing the aligned reads.

### Coverage signal extraction and denoising
Each cfDNA BAM file was segmented into 1000 bp-long non-overlapping bins, and the number of alignments in each bin was reported using `bedtools intersect v2.30.0`. Only properly paired non-duplicated reads with high mapping quality (MAPQ > 30) were considered. These genome-wide counts represent the raw coverage signal, which was then brought into a state appropriate for statistical analysis through several pre-processing steps, as follows. First, we subtracted from the raw signal all ENCODE blacklisted regions v2 and all difficult regions from the Genome-In-A-Bottle project. In a subsequent filtering step, we removed bins with mappability score less than 50%, as well as bins with excessively high or low GC content, as these indicate the presence of potential artefacts. In the following step, we normalised the coverage signal in each bin by dividing by the genome-wide median coverage, and then taking the $\log_2$ value of the resulting ratio. Each bin was then annotated with its median GC content and mappability score. Since the coverage signal in each bin is dependent on its GC content and mappability, it must be corrected for any bias introduced by these two variables. For this purpose, we used a Generalised Additive Model (GAM) to describe the coverage signal as a function of GC content and mappability. The choice of a GAM allows us to model the coverage without assuming any particular form for the functions that model its dependence on GC content and mappability. In `R v4.1.3`, this is done using the function `gam` from package `mgcv v1.8-40`, as follows: `gam(y ~ s(GC, bs='cs') + s(MAP, bs='cs'))`. In this code snippet, $y$ is the vector of normalised coverage values across all bins, while GC

and MAP are the corresponding vectors of GC content and mappability scores. The coverage $y$ is corrected by subtracting its dependencies on GC and MAP, as estimated using the above procedure. Finally, the de-biased coverage signal in each bin is normalised again by subtracting the genome-wide average. The filtered and de-biased coverage for all plasma cfDNA samples obtained from the previous steps is further denoised, using the identically pre-processed non-cancer cfDNA samples from the SCAN pathway for characterising the systematic background noise. These samples encapsulate unwanted variance, which we wish to remove from all cancer samples. This is achieved in two stages. We start by collecting the non-cancer SCAN samples in a matrix $M$ with $m$ columns (equal to the number of samples) and $n$ rows (equal to the total number of bins), where $m$ is typically much smaller than $n$. We factorise this matrix using its singular value decomposition, $M = U\Sigma V^T$, where $\Sigma$ is an $m \times m$ square matrix with the singular values along its main diagonal in decreasing order, $U$ is an orthogonal $n \times m$ matrix with the corresponding left singular vectors as its columns, and $V$ is also an orthogonal $m \times m$ matrix with the corresponding right singular vectors as its columns. If the coverage signal for a particular sample is $y$, the systematic background noise is given by the product $UU^T y$, which is subsequently subtracted from $y$. In the second stage, the coverage signal is further denoised through application of the Savitzky-Golay filter[49], a digital filter that smooths the data without distorting the underlying signal. In `R v4.1.3`, this can be achieved using the function `sgolayfilt` from the R package `signal v0.7-7`. We use a filter of order 3 and length 1000 bins ($10^6$ base pairs). After smoothing, the denoised signal is again normalised by subtracting the genome-wide median.

### Methylation calling
TAPS induces specific modification signatures that are essential for methylation analysis. Methylated cytosines undergo a C>T conversion on the forward strand and a G>A conversion on the reverse strand, while unmethylated cytosines remain unchanged. This strand-specific signature is critical for accurate methylation detection. The open-source TAPS methylation caller, `asTair v3.3.2`, was used to call methylation from the cfDNA and the germline BAM files. The `asTair` algorithm recognizes TAPS changes (mC>T) and differentiates these from C>T variants or variants resulting in a methylated C>T. It can also indicate the co-location of methylation change and a variant. The aligned reads were processed to detect methylated cytosines using the `call` command of `asTair`. This step involved several sub-processes: 1) Base Quality Filtering: bases with a quality score below 30 were excluded to ensure data reliability. 2) Context Identification: cytosine bases within specified sequence contexts (e.g., CpG) were identified. 3) Conversion Detection: methylation was inferred from C>T conversions on the forward strand and G>A on the reverse strand. 4) Strand-Specific Analysis: it ensures that modifications appeared only on the expected strand, as TAPS modifications are strand-specific. To distinguish true methylation events from sequencing artifacts or variants, the support for modifications on both strands was assessed. Specifically, C>T modifications should only appear on the forward strand. Observations of C>T modifications on the reverse strand indicate potential non-TAPS variants. For example, if the C>T ratio on the forward strand suggests a modification ratio of 0.25 and there is no support on the reverse strand, this implies a TAPS-like modification. If the same modification ratio is observed on both strands, this suggests a variant, rather than methylation. Positions with both variation and methylation can be identified by differing frequencies of support on each strand.

### Somatic variant calling
Upon `asTair` analysis, a cfDNA and a germline methylation VCF was generated for each patient. For both the cfDNA and germline, the TAPS methylation calls were filtered to keep positions indicating methylation alone as these are then used to filter out methylation from cfDNA variant calls in a later step. The `GATK4 HaplotypeCaller` was used to call

germline variants from germline BAM files with `GATK` recommended settings. `bcftools mpileup v1.15` was used to generate a pileup file from the cfDNA BAM file. To identify potential variants in the cfDNA sample an extensive filtering process was applied to the pile-up. Specifically, the raw cfDNA variant calls were sorted, decomposed and normalized using `bcftools` to ensure consistency and improve downstream analysis accuracy when filtering germline variants from this set. Normalised variant VCF was further sorted and indexed. Germline variants (Haplotype Caller output VCF) and germline methylation calls (filtered `asTair` output) were filtered out if present in cfDNA variants. From this, cfDNA methylation calls (filtered `asTair` output) were then filtered out. The variants obtained from the previous step underwent quality filtering based on established criteria, such as read depth, variant allele frequency, quality scores, and annotation information, to ensure the reliability and accuracy of the final variant call set. For SNVs, exclusion criteria were as follows: DP > 200 or DP < 50, MQ < 60, MQBZ < −9, RPBZ < -5 or RPBZ > 5, MQ0F > 0, BQBZ < −4, VAF > 0.3 or VAF < 0.03. For INDELs, the following exclusion criteria were used: DP > 200 or DP < 50, VAF > 0.3 or VAF < 0.03. The resulting cfDNA VCF was annotated using `VEP` and converted to MAF format for easier use of annotations. Finally, the non-cancer SCAN samples were used for denoising, by generating a merged VCF of all these samples, which was used to remove shared variants from the cfDNA VCF files for all samples (Supplementary Fig. 2).

## Fragment-based methylation analysis

Lists of differentially methylated CpGs identified in TCGA were downloaded via `SMART` for the cancer types in this study. These lists were used to generate BED files of differentially methylated regions by extending the coordinates 100 bp in each direction and merging any of the resulting regions that overlapped. Any regions that contained fewer than 3 differentially methylated CpGs after this process were discarded along with hypomethylated regions, leaving 2113 hypermethylated regions from 5 TCGA studies: COAD, ESCA, KIRC, KIRP and PAAD (Supplementary Data 2). We chose to focus on hypermethylation markers, where baseline methylation should be close to zero in non-cancer individuals and only successful conversion by the TAPS chemistry would result in a positive signal. For each sample, any aligned reads that overlapped with the selected methylation marker regions were matched with their mate pair and the entire fragment was considered together. Fragments that overlapped with fewer than three CpGs in any region were discarded and then methylation calling was performed based on TAPS base changes. Fragments were then classified as either originating from a tumour or originating from healthy cells based on the proportion of callable positions that were modified by the TAPS chemistry. For the purposes of this study the threshold for classification as a tumour fragment was set at 80% modification. The rationale behind this threshold was to improve sensitivity (allowing for fragments where not all CpG positions were converted; for example, if methylated bases were replaced during end-repair or the chemistry was not 100% efficient), whilst not sacrificing the specificity of the fragment-based approach by ensuring that only well converted fragments were considered as originating from tumour cells (i.e., fragments with fewer than 5 CpGs must be fully converted; fragments with between 5 and 9 CpGs can have only one CpG unconverted; fragments with between 10 and 14 CpGs can have at most two CpGs unconverted, and so on). Finally, the proportion of fragments classified as originating from tumour cells was calculated for each marker. Samples from the SCAN cohort were used to identify markers that were not specific to cancer. Any marker overlapping with one or more tumour-originating fragments in these samples was excluded, leaving 377 markers for downstream analysis of the cancer cases and non-cancer CBS controls (Supplementary Data 2). Among these markers, 280, 51, 42 and 4 were specific to colorectal, oesophageal, pancreatic and renal cancer, respectively.

## Statistical analysis for cancer detection

The clean SNV/INDELs, methylation and coverage signals from each cancer plasma sample were compared to the corresponding signals from non-cancer plasma CBS samples. In the case of the coverage data, the signal was first aggregated in each chromosome arm in each sample. The mean and standard deviation of the total coverage signal for each chromosome arm were calculated across all non-cancer controls. These statistics were subsequently used in the calculation of coverage z-scores for each chromosome arm in each cancer sample. For a particular cancer sample, the z-score at chromosome arm $k$ is given by $z_k = (y_k - m_k)/s_k$, where $y_k$ is the aggregated coverage signal at arm $k$ in the cancer sample, and $m_k$ and $s_k$ are the mean and standard deviation of the aggregated coverage signal at chromosome arm $k$ across the non-cancer CBS controls. The sampling distribution of the statistic $t_k = z_k/\sqrt{1+N^{-1}}$ is a t-distribution with $N-1$ degrees of freedom, where $N$ is the number of non-cancer CBS controls used for the calculation of $m_k$ and $s_k$. From this, we calculated two-sided p-values $p_k$, which were corrected for multiplicity using the Benjamini-Hochberg procedure. In each cancer sample, those arms with corrected p-values lower than 5% are copy number aberrant in comparison to the corresponding arms of the non-cancer controls. To derive a sample- and modality-specific p-value, we selected the aberrant arms, and we combined their p-values using Stouffer's method: given uncorrected p-values $p_k$, we derived scores $z_k = \Phi^{-1}(1-p_k)$, where $\Phi^{-1}$ is the inverse cumulative distribution function of the standard normal distribution. These are then combined into an overall z-score $z_m = \sum_k z_k/\sqrt{K}$, where $K$ is the total number of aberrant chromosome arms. Since $z_m$ follows a standard normal distribution, a sample-specific p-value for this data modality was derived as $p_m = 1 - \Phi(z_m)$. The cancer samples with p-values less than 5% are copy-number aberrant in comparison to the non-cancer controls. Sample-specific p-values were also derived from the SNV/INDEL and methylation data using the same methodology. In the case of SNVs/INDELs, we used the $\log_{10}$ of the mutation burden of each chromosome arm. For the methylation data, we used the logit of the total methylation ratio in each of the 377 methylation markers, derived as explained above. For each sample, a p-value across all three modalities was derived using again Stouffer's method. If weights were available quantifying our confidence in each data modality, a weighted sample-specific multi-modal z-score would take the form $z = \sum_m w_m z_m/\sqrt{\sum_m w_m^2}$, where index $m$ enumerates each data modality. Again, this is distributed according to the standard normal distribution, which allows easy calculation of an associated p-value, as explained above.

We adopted Stouffer's method because of its straightforward applicability to our problem, the ease of incorporating weights (if such weights become available), and the easy adaptability of the method to the absence of one or more data modalities from one or more patients (which is not uncommon in a real-life clinical scenario).

An alternative method for integrating data modalities is Fisher's method. Given modality-specific p-values $p_m$ in a particular sample (which were derived as explained above), we calculate the statistic $S = -2\sum_m \log p_m$. This statistic follows the $\chi^2$ distribution with $2M$ degrees of freedom, where $M$ is the number of data modalities (in our case, $M = 3$).

In addition, we used three common algorithms in statistics and machine learning, as an alternative approach for integrating different data modalities, namely Logistic Regression (using function `glm` in `R v4.1.3` with argument `family=binomial(link='logit')`), Random Forest (using function `randomForrest` in the R package `randomForrest v4.7.1`) and Support Vector Machine (using function `svm` in the R package `e1071 v1.7`). In all three cases, we used the modality-specific scores $z_m$ as predictors in a binary classification model, where the response variable was the clinical status of each patient (i.e., cancer or non-cancer). Algorithm parameters were fixed at their default settings in all three cases.

## Prediction of cancer origin

Finally, we developed multi-cancer classifiers using the Random Forest and Support Vector Machine paradigms, as above, as well as the Penalised Multinomial Regression formalism, as implemented in the R package `glmnet v4.1-7` with argument `family='multinomial'`. In all three cases, we aimed to predict the cancer type/origin using as predictors the coverage signal, the $\log_{10}$-transformed number of somatic SNVs/INDELs across each chromosome arm, as well as the logit of the methylation ratio across 377 TCGA regions (see Methods section for each data modality, above). Algorithm parameters were fixed at their default values for Random Forest and Support Vector Machine, but for `glmnet`, the L1 penalty was estimated using leave-one-out cross-validation with function `cv.glmnet` (see relevant section below for details).

## Calculation of ctDNA fractions

In those samples with copy-number aberrations, we were able to calculate the ctDNA fraction based on further analysis of the coverage data. First, the filtered, debiased, denoised and normalised coverage signal was divided into contiguous segments of relatively constant coverage. Genome segmentation was conducted using `DNAcopy v1.68.0` after subsampling the coverage signal every 1000 bins ($10^6$ base pairs). Segments less than 3 standard deviations apart were merged into a single segment. Assuming the $\log_2$-transformed coverage at bin $i$ in segment $k$ is $y_{ki}$, we calculated the untransformed coverage $x_{ki} = 2^{y_{ki}+1}$. Typically, the plasma cfDNA is a mixture of DNA fragments originating from normal diploid cells and cancer cells. Previous approaches make the strong assumption that the tumour component of the plasma cfDNA originates from one major clone and one subclone. Here, we make the less restrictive assumption that the ctDNA originates from an unspecified number (one or more) of clones of not-necessarily-diploid cancer cells. It follows that the expected value of $x_{ki}$ is equal to $2(1-\rho) + \rho\bar{c}_k$, where $\rho$ is the ctDNA fraction and $\bar{c}_k$ is the average ploidy at segment $k$ across all cancer cells. It is important to notice that $\bar{c}_k$ is, in general, not an integer, unless the copy number aberration overlapping segment $k$ is a clonal event, i.e., harboured by all cancer cells. We model $x_{ki}$ using a normal distribution, as follows:

$$x_{ki} \sim Normal\left(2(1-\rho) + \rho\bar{c}_k, s_k^2\right) \tag{1}$$

where $s_k^2 = \sum_{i=1}^{N_k}(x_{ki} - m_k)^2/(N_k - 1)$ is the observed variance of $x_{ki}$, $m_k = \sum_{i=1}^{N_k} x_{ki}/N_k$ its observed mean, and $N_k$ the number of bins supporting segment $k$. Since $\bar{c}_k$ is not an integer, we can impose a uniform prior on it between 0 and a maximum ploidy value, e.g., 4. Similarly, we impose a uniform prior on $\rho$ between 0 and 1. The advantage of this formulation is that it allows for a very efficient Gibbs sampling inference scheme for calculating the posterior distribution of $\bar{c}_k$ (across all segments) and $\rho$. This scheme consists of repeatedly sampling from the following conditional posteriors for $\rho$ and $\bar{c}_k$:

$$\rho \sim Normal\left(\frac{\sum_{k=1}^{K} N_k(m_k - 2)(\bar{c}_k - 2)/s_k^2}{\sum_{k=1}^{K} N_k(\bar{c}_k - 2)^2/s_k^2}, \frac{1}{\sum_{k=1}^{K} N_k(\bar{c}_k - 2)^2/s_k^2}\right) \tag{2}$$

$$\bar{c}_k \sim Normal\left(\frac{m_k - 2(1-\rho)}{\rho}, \frac{s_k^2}{N_k \rho^2}\right) \tag{3}$$

When applied to a particular sample, we ran the above procedure for 10 K iterations, and we recorded the mean and variance of the resulting sample chains, after they had attained equilibrium (usually after ~5 K iterations).

## In silico data generation for methods validation

To validate the above pipeline, we simulated non-cancer and cancer plasma samples at various ctDNA fractions using actual plasma samples as templates. For each in silico sample, we simulated aggregate coverage and somatic mutation burden per chromosome arm, and methylation ratios per methylation marker. The aggregate coverage signal in arm $k$ for a non-cancer sample was simulated by sampling a point from a normal distribution with mean $m_k$ and variance $s_k$. These statistics were estimated from the actual non-cancer CBS samples. The aggregate coverage signal in arm $k$ for a cancer sample was simulated as a mixture: $y_k = wx_k^t + (1-w)x_k^h$, where $x_k^h$ is a random point from a normal distribution with mean $m_k$ and standard deviation $s_k$, $x_k^t$ is the aggregate coverage signal in chromosome arm $k$ from an actual tumour sample with ctDNA fraction $\rho$, and $w$ is the ratio $r/\rho$, where $r$ is the target ctDNA fraction of the simulated cancer plasma sample. Mutation burden and methylation signals were generated in the same way using respectively the arm-specific mutation burden and methylation marker-specific statistics in place of $m_k$ and $s_k$ calculated from the non-cancer CBS controls. For all these mixtures, we used a colorectal cancer plasma sample with $\rho = 9\%$. For each of ten target ctDNA fractions spanning the range from 0.1% to 2%, we simulated 1000 non-cancer and 1000 cancer samples, resulting in a total of 20 K simulated plasma samples. The capacity of the previously described statistical methodology to discriminate between cancer and non-cancer plasma samples at decreasing ctDNA fractions was assessed using these simulated data and the area under the ROC curve (AUC) as performance metric. For recent publications using synthetic data in liquid biopsy research, we refer the reader to[21,26,50,51].

## Leave-one-out cross-validation (LOO-CV)

Furthermore, we adopted a LOO-CV approach to estimate the generalisation capacity of the above methodologies. In summary, for each tested model (Stouffer's and Fisher's methods, Logistic Regression, Random Forrest, Support Vector Machine, and Penalised Multinomial Regression), we removed the first subject from our cohort, trained the model on the remaining subjects, and predicted the cancer status of the first subject using the trained model. Since the first subject was not used in training, it essentially constitutes an independent sample. We repeated the same steps with all subsequent subjects until all were used for prediction. This procedure estimates the out-of-sample (or generalisation) error of each model on an effectively independent cohort, which is the same size as our original dataset. In the case of Penalised Multinomial Regression, at each iteration of the LOO-CV, the L1 penalty was estimated in a second (inner) loop of LOO-CV on the training data only using the `glmnet` function `cv.glmnet`. In all other models, parameters were fixed at their default values. Finally, to account for class imbalance (i.e., the fact that not all cancer groups in our cohort are of equal size), at each LOO-CV iteration, we weighted each case by a weight proportional to $\max(n_1, n_2, \ldots)/n$, where $n$ is the number of subjects in the diagnostic group of the case in the training data at each iteration. $\max(n_1, n_2, \ldots)$ is the number of subjects in the largest diagnostic group, i.e. the number of patients with colorectal cancer in the training data at each iteration of the LOO-CV procedure.

## Reporting summary

Further information on research design is available in the Nature Portfolio Reporting Summary linked to this article.

# Data availability

TAPS data (in the form of BAM files) from 214 samples generated in this study (plasma and matched germline pairs from 91 non-cancer and cancer subjects, matched fresh-frozen tumour biopsies from 16 subjects among these, as well as several follow-up plasma samples from 10 subjects with colorectal cancer) has been deposited in the European

Genome-Phenome Archive (EGA) under the study with accession code EGAS50000000715. Source data is provided as a source data file. Source data are provided with this paper.

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

## Acknowledgements

This research was funded by Innovate UK (Competition: Application of whole genome sequencing approaches to cancer; Project title: Base Genomics: A novel method for single-step, ultra-sensitive, combined DNA methylation and mutation detection of cancer from liquid biopsies using WGS; Grant application: 49382) and the National Institute for Health Research (NIHR) Oxford Biomedical Research Centre (BRC4 Cancer theme awarded to AS). We further acknowledge the contribution to this study made by the Oxford Centre for Histopathology Research and the Oxford Radcliffe Biobank, which are supported by the University of Oxford, the Oxford CRUK Cancer Centre and the NIHR Oxford Biomedical Research Centre (Molecular Diagnostics Theme/Multimodal Pathology Subtheme), and the NIHR CRN Thames Valley network. BDN is supported by a National Institute of Health Research Academic Clinical Lectureship, a CRUK Research Careers Committee Postdoctoral Fellowship (RCCPDF\100005), and he is the Early Detection Theme Lead for the CRUK Oxford Cancer Centre (CTRQQR-2021\100002). The views expressed are those of the authors and not necessarily those of the NHS, the NIHR or the Department of Health and Social Care.

## Author contributions

A.S. conceptualised the study, raised funds to implement the study, and directed the study. A.S., D.V.V., A.C., C.T. and H.D. designed the study. C.T., H.D. oversaw laboratory operations. A.C., J.B., F.M.S., T.S., B.M., Y.B.L. and C.X.S. designed and performed experiments. S.A. and B.D.N. provided clinical data and samples; D.V.V., A.A. and K.H. conducted coverage and CNV analysis; D.V.V., J.B., A.A., and A.R. conducted methylation analysis. D.V.V., N.T., R.S. and A.R. conducted somatic SNV analysis. D.V.V. conducted integrative statistical analysis and developed machine learning classifiers. D.V.V., A.S. and B.D.N. interpreted the results and wrote the manuscript with input from all authors.

## Competing interests

A.S. is on the advisory board for Janssen and BeiGene and is the Founder of SERENOx. She also receives research funding from AstraZeneca and Janssen and in-kind contributions from Illumina and Oxford Nanopore Technologies. BDN has received research funding from GRAIL and is an unpaid member of the GRAIL Clinical Advisory Group. D.V.V., A.C. and H.D. own shares in SerenOx. N.T., J.B., R.S., A.R., T.S., B.M., Y.B.L. and C.T. were employees at Exact Sciences Innovation LTD for the duration of the study. The remaining authors declare no competing interests.
