## [Transparent Peer Review file · Nature Communications]

Multimodal cell-free DNA whole-genome TAPS is sensitive and re-veals specific cancer signals

Corresponding Author: Dr Dimitrios Vavoulis

Version 0:

Reviewer comments:

Reviewer #1

(Remarks to the Author)

Dear Authors,

I have thoroughly reviewed the manuscript and offer the following comments:

Feasibility of Experimental Methods: The proposed TAPS sequencing method for ctDNA detection, with its base-level resolution of 5-methylcytosines and 5-hydroxymethylcytosines, shows promise. While the feasibility of the technique is validated on real patient samples, further large-scale and multicenter validation is necessary to confirm its effectiveness in multi-cancer early detection. I suggest that the manuscript could delve into more specific details regarding the methodology employed for methylation analysis to enhance reader comprehension.

Integration of Multi-Modal Information: The integration of information from various modalities, including copy number aberrations, single nucleotide variants, insertions/deletions, and methylation signals, offers a comprehensive analysis for ctDNA detection. However, a more in-depth discussion is needed to elucidate the specific advantages and clinical utility of this approach across different cancer types and clinical scenarios. Additionally, a detailed description of the methylation analysis methodology would improve clarity and reproducibility.

Clinical Application Prospects: While the manuscript suggests potential clinical applications for early detection in various cancers, such as colorectal, esophageal, pancreatic, renal, ovarian, and breast cancers, further research support and practical validation are required. Discussing the potential implications of the findings for clinical practice and patient care would add value.

Design of Validation Experiments: The validation experiments, relying on synthetic data, evaluate the discriminatory ability of the method. However, given the heterogeneity and complexity of real clinical samples, real-world validation is crucial for reliability and generalizability. The authors should address their plans for overcoming challenges associated with real-world validation and acknowledge the complexities involved.

In conclusion, this manuscript introduces a potentially valuable ctDNA detection method, incorporating multi-modal information for cancer early detection. Further experimental data and clinical validation are necessary to support its practical application. I appreciate the authors' consideration of these suggestions in their discussion and encourage them to delve deeper into the specific aspects highlighted, particularly providing more specifics on the methylation analysis methodology.

Reviewer #3

(Remarks to the Author)

The authors reported a new analytical pipeline to detect cancer-specific signals from cfDNA using TAPS with high coverage WGS (80x). The study claimed that this analytical pipeline will help separate cancer patients from those without cancer and detect minimal residual disease. Overall, the study is interesting and provides a novel approach to the emerging field of liquid biopsy-based cancer diagnosis. However, the study is highly descriptive and lacks rigorous validation in additional patient cohorts. The followings are the main concerns that need to be addressed.

1. Small study cohort size (N=60) is one of the major concerns. Although simulation data showed high AUC, the new algorithm needs an additional patient cohort for validation.
2. What was the rationale behind using Stouffer's method? Did the authors also test other similar methods that can be used to validate similar outcomes/predictions?
3. It is not clear how to convert integrated p-values to integrated scores. Please explain.
4. It would be great if the authors could demonstrate comparisons between the new algorithm and other published algorithms in the same patient cohort.

5. The authors showed potential use of the integrated ctDNA score for detection of MRD and monitoring of treatment responses. Although the ctDNA scores in some cases were in good agreement with clinical outcomes, the results were based on a few cases and limited timepoints of blood collection. Longitudinal analysis with more blood collection times is needed.
6. Due to high cost, this new approach that applies WGS with 80 x coverage may not be suitable for future clinical applications. Pairing with germline samples will further increase the overall cost. Other low-cost options could be a better option.
7. The study generated a huge amount of data. However, there is no data-sharing information. The sequencing data should be deposited in the public domain for downloading.
8. The manuscript should provide supplementary data to support results. For example, mutational data from all detected mutations to filtered mutations along their allele frequencies and sequencing depth. Also, information on cancer-associated methylation fragments needs to be included as supplementary materials.
9. Line 477: "For the purposes of this study the threshold for classification as a tumour fragment was set at 80% modification." What is the rationale to use the 80% cutoff?
10. What does "non-coding missense mutations" in lines 192-193 mean? Only protein-coding regions have missense mutations.
11. The current algorithm lacks the capacity to classify tissue of cancer origin, which is essential in the current standard of multi-cancer detection.
12. Lines 177-182 described 4 filtering criteria including d) further denoised moving all variants shared with any of 21 age-matched non-cancer plasma samples from patients. It is unclear if the somatic mutations reported in the paragraph starting from line 184 also went through the filtering criteria. If yes, the control plasma should have 0 somatic mutations. However, it reported 125-192 mutations in control samples.
13. It is unclear why cancer patients experienced more passenger mutations. Since these mutations are not cancer-driven, cancer patients may have similar passenger mutation frequency as controls, which needs further clarification.
14. Please clarify why the study uses 21 non-cancer controls for data normalization and filtering but uses 9 non-cancer controls for case-control classification. Are the 9 controls from the 21 controls?

Version 1:

Reviewer comments:

Reviewer #3

(Remarks to the Author)

Thank the authors for their effort in revising the manuscript. The new version of the manuscript has addressed most of my concerns and comments. Although not required, I am wondering if the technology and algorithm can capture cancer (and cancer cell type) signals by downsizing sequencing coverage from 80x to 10x or even 1x. I think that the high cost of current 80x coverage is a limiting factor for further application of this technology in clinic. I am also wonder if authors can convert the TAPS-based methylation to bisulfite-based methylation format. In doing so, authors can use existing deconvolution tool for cell type deconvolution.

(Remarks on code availability)

Reviewer #4

(Remarks to the Author)

The authors have adequately addressed Reviewer #1's comments. I have no further comments on the revised manuscript.

(Remarks on code availability)

RESPONSE TO REVIEWERS' COMMENTS

Reviewer 1: expertise in ctDNA detection methodology

Dear Authors,

I have thoroughly reviewed the manuscript and offer the following comments:

We would like to thank the reviewer for reading our paper and for his/her useful comments.

1. Feasibility of Experimental Methods: The proposed TAPS sequencing method for ctDNA detection, with its base-level resolution of 5-methylcytosines and 5-hydroxymethylcytosines, shows promise. While the feasibility of the technique is validated on real patient samples, further large-scale and multicenter validation is necessary to confirm its effectiveness in multi-cancer early detection.

We would like to thank the reviewer for raising this point. The aim of this retrospective cohort study was to demonstrate the potential sensitivity of our technology in a limited set of samples from patients with different types and stages of cancer. We agree with the reviewer that the next step would be to validate these promising results in a prospective randomised multi-centre study in high-risk populations. We mention the need for further clinical validation in paragraphs 5 and 6 of the Discussion in the revised manuscript, and we are currently actively engaging with the University's Business Unit to seek the funds that would allow us to transition to a Phase 3 study.

2. I suggest that the manuscript could delve into more specific details regarding the methodology employed for methylation analysis to enhance reader comprehension.

Following the reviewer's comment, we have added the subsections **Read Alignment** and **Methylation Calling** in the Methods section of the paper. For the methylation calling, we write:

Methylation calling: TAPS induces specific modification signatures that are essential for methylation analysis. Methylated cytosines undergo a C>T conversion on the forward strand and a G>A conversion on the reverse strand, while unmethylated cytosines remain unchanged. This strand-specific signature is critical for accurate methylation detection. The open-source TAPS methylation caller, asTair v3.3.2, was used to call methylation from the cfDNA and the germline BAM files. The asTair algorithm recognizes TAPS changes (mC>T) and differentiates these from C>T variants or variants resulting in a methylated C>T. It can also indicate the co-location of methylation change and a variant. The aligned reads were processed to detect methylated cytosines using the call command of asTair. This step involved several sub-processes: 1) **Base Quality Filtering:** bases with a quality score below 30 were excluded to ensure data reliability. 2) **Context Identification:** cytosine bases within specified sequence contexts (e.g., CpG) were identified. 3) **Conversion Detection:** methylation was inferred from C>T conversions on the forward strand and G>A on the reverse strand. 4) **Strand-Specific Analysis:** it ensures that modifications appeared only on the expected strand,

as TAPS modifications are strand-specific. To distinguish true methylation events from sequencing artifacts or variants, the support for modifications on both strands was assessed. Specifically, C>T modifications should only appear on the forward strand. Observations of C>T modifications on the reverse strand indicate potential non-TAPS variants. For example, if the C>T ratio on the forward strand suggests a modification ratio of 0.25 and there is no support on the reverse strand, this implies a TAPS-like modification. If the same modification ratio is observed on both strands, this suggests a variant, rather than methylation. Positions with both variation and methylation can be identified by differing frequencies of support on each strand.

3. Integration of Multi-Modal Information: The integration of information from various modalities, including copy number aberrations, single nucleotide variants, insertions/deletions, and methylation signals, offers a comprehensive analysis for ctDNA detection. However, a more in-depth discussion is needed to elucidate the specific advantages and clinical utility of this approach across different cancer types and clinical scenarios.

While future validation in larger data sets of different cancer types and stages is needed, we provide proof-of-principle that deep whole genome sequencing combining depth with breadth of sequencing to allow integration of information from various modalities (copy number aberrations, single nucleotide variants, insertions/deletions, and methylation signals) is highly sensitive.

Circulating tumour DNA is highly fragmented, of low abundance and highly diluted in circulating germline DNA originating from peripheral blood mononuclear cells. In addition, some cancers might not have many copy number changes. Instead, they display a characteristic acquired mutational or epigenetic signature profile. A main advantage of our approach is therefore that it considers this cancer heterogeneity by using different types of aberrant signals that are enriched for somatic (acquired) events to define the probability for a cancer diagnosis.

Following the reviewer's comment, we highlight the above in paragraphs 3 and 4 of the Discussion in the revised manuscript.

4. Additionally, a detailed description of the methylation analysis methodology would improve clarity and reproducibility.

Please, see our response to comment 2.

5. Clinical Application Prospects: While the manuscript suggests potential clinical applications for early detection in various cancers, such as colorectal, oesophageal, pancreatic, renal, ovarian, and breast cancers, further research support and practical validation are required. Discussing the potential implications of the findings for clinical practice and patient care would add value.

The results of our study presented here will pave the way to a future larger scale validation in people with high-risk features. These could be defined by age, non-specific symptoms or genetic cancer predisposition, life-style habits, or the refusal to participate in any of the existing more invasive screening procedures for cervical, breast, lung or colorectal cancer. Past and ongoing studies led by others have already highlighted some of the practical challenges associated with early cancer detection from liquid biopsies, particularly the potential psychological, emotional and financial “toxicity” from false positive results. For MCED tests to enter clinical practice, future efforts are needed to further improve test attributes, especially positive predictive values. Importantly, long-term follow-up of the cohort study Detect-A already suggests that overall survival of patients correctly identified with cancer from liquid biopsy screening is improved while emotional or physical morbidity from false positive results recovers soon after testing. Additional and prospective data will be hopefully available from longer follow-up of the GRAIL-NHS study.

Following the reviewer’s comment, we highlight the above in paragraph 6 of the Discussion in the revised manuscript.

6. Design of Validation Experiments: The validation experiments, relying on synthetic data, evaluate the discriminatory ability of the method. However, given the heterogeneity and complexity of real clinical samples, real-world validation is crucial for reliability and generalizability.

We recognise the need for external validation on real clinical samples, as pointed out by the reviewer. Since additional sequencing is currently not possible (due to cost and time-constraints), our first course of action was to use high-fidelity synthetic data for validation. This is a reasonable approach, which has been adopted in recent publications in reputable journals, for example, Zviran et al. *Nature Medicine* 2020; Mathios et al. *Nature Communications* 2021; Yu et al. *Genome Medicine* 2024; Widman et al. *Nature Medicine* 2024. We were cautious to use actual biological samples as templates in our simulations, thus generating synthetic data that closely mimic the variability of the actual data.

Furthermore, following the reviewer’s comment, we adopted a cross-validation approach (leave-one-out cross-validation or LOO-CV) to estimate the generalisation capacity of our methodology. In summary, we removed the first subject from our cohort, trained our models on the remaining subjects, and predicted the cancer status of the first subject using the trained models. Since the first subject was not used in training, they essentially constitute an independent sample. We repeated the same steps with all subsequent subjects until all were used for prediction. This procedure estimates the out-of-sample (or generalisation) error of the methodology on an effectively independent cohort, which is the same size as our original dataset. The results from this type of analysis are summarised in Supplementary Figure S4. The AUC when using all data modalities for classification was 83.5%. Furthermore, we tested additional approaches for integrating modality-specific scores (Supplementary Figure S5), as

well as multi-class classifiers for predicting cancer origin (Supplementary Figure S6), all of which were also validated using LOO-CV.

We have now revised the manuscript to highlight the above points (see updated Results section ***Integration of multiple genomic modalities for ctDNA detection***, Supplementary Figures S4, S5 and S6, as well as the new Methods section ***Leave-One-Out Cross-Validation***). Also, we have added the citations mentioned above.

7. The authors should address their plans for overcoming challenges associated with real-world validation and acknowledge the complexities involved.

As mentioned above, the results presented here must be validated in randomised controlled studies of people at risk before implementation into public healthcare systems that rely on re-imburement from national or private insurance contributions. Once randomisation demonstrates clinical utility and cost-effectiveness, a complementary and important step in the validation process is the post-approval real-world (RW) data collection (Phase 4). Although data is often of lesser quality and incomplete compared to formal prospective studies, RW data will reveal important implementation challenges, for example: who the test is prescribed for (at risk individuals? What type of at-risk individuals?); How often is the test prescribed? How are false positive results handled in the RW?

Following the reviewer's comment, we have added paragraph 7 in the Discussion of the revised manuscript.

7. In conclusion, this manuscript introduces a potentially valuable ctDNA detection method, incorporating multi-modal information for cancer early detection. Further experimental data and clinical validation are necessary to support its practical application. I appreciate the authors' consideration of these suggestions in their discussion and encourage them to delve deeper into the specific aspects highlighted, particularly providing more specifics on the methylation analysis methodology.

We hope that our responses satisfy the points raised by the reviewer. Once more, we would like to thank them for their useful comments, which helped us improve this manuscript.

Reviewer 3: expertise in ctDNA bioinformatics

The authors reported a new analytical pipeline to detect cancer-specific signals from cfDNA using TAPS with high coverage WGS (80x). The study claimed that this analytical pipeline will help separate cancer patients from those without cancer and detect minimal residual disease. Overall, the study is interesting and provides a novel approach to the emerging field of liquid biopsy-based cancer diagnosis. However, the study is highly descriptive and lacks rigorous validation in additional patient cohorts. The followings are the main concerns that need to be addressed.

We would like to thank this reviewer for their time and for their valuable comments.

1. Small study cohort size (N=60) is one of the major concerns. Although simulation data showed high AUC, the new algorithm needs an additional patient cohort for validation.

In this study, we recruited 61 cancer and 30 non-cancer subjects (21 from the Oxford SCAN pathway and 9 from Cambridge BioScience; see also our response to point 14 below). From each subject, we collected a plasma and a germline sample. From 16 of these subjects, we also collected fresh-frozen biopsy samples, and from another 10 of these subjects we collected one or more follow-up plasma samples. In total, we generated WGS data from 214 samples (germlines at 30x; all other samples at 80x). This is the largest and most deeply sequenced TAPS dataset currently available, and we are confident that it will be a valuable resource in the cfDNA research community.

Nevertheless, as the reviewer points out, we recognise the need for independent validation of our methodology. Since additional sequencing is currently not possible (due to cost and time constraints), our first course of action was to use high-fidelity synthetic data for validation. This is a reasonable approach, which has been adopted in recent publications in reputable journals, for example, Zviran et al. *Nature Medicine* 2020; Mathios et al. *Nature Communications* 2021; Yu et al. *Genome Medicine* 2024; Widman et al. *Nature Medicine* 2024. We were cautious to use actual biological samples as templates in our simulations, thus generating synthetic data that closely mimic the variability of the actual data.

Furthermore, following the reviewer's comment, we adopted a cross-validation approach (specifically, leave-one-out cross-validation or LOO-CV) to estimate the generalisation capacity of our methodology. In summary, we removed the first subject from our cohort, trained our model on the remaining subjects, and predicted the cancer status of the first subject using the trained model. Since the first subject was not used in training, they essentially constitute an independent sample. We repeated the same steps with all subsequent subjects until all were used for prediction. This procedure estimates the out-of-sample (or generalisation) error of the methodology on an effectively independent cohort, which is the same size as our original dataset. The results from this type of analysis are summarised in Supplementary Figure S4. The cross-validated sensitivity, specificity and AUC when using all data modalities for classification were 85.2%, 88.8% and 83.5%, respectively.

We have now revised the Results section **Integration of multiple genomic modalities for ctDNA detection**, we have added the methodology section **Leave-One-out Cross-Validation**, and we have added the citations mentioned above.

2. What was the rationale behind using Stouffer's method? Did the authors also test other similar methods that can be used to validate similar outcomes/predictions?

We use Stouffer's method in two different instances in our analysis. First, to integrate arm- or region-specific p-values into a modality-specific p-value for a particular patient. Second, to integrate all modality-specific p-values for a particular patient into an overall (multi-modal) p-value, which we use to reject the null hypothesis that the plasma of the patient does not include a detectable amount of ctDNA. In both cases, the individual p-values constitute independent pieces of evidence, which need to be integrated into a final p-value that can be used in clinical decision making (i.e., whether the patient has cancer or not).

This problem is the subject of an area of Statistics known as "meta-analysis", which includes Stouffer's method as one of its most popular approaches. The way Stouffer's method works is the following: given independent p-values p_1, p_2, p_3, \dots we calculate z-scores z_1, z_2, z_3, \dots , where each score, z_i , is given by the inverse cumulative distribution function Φ^{-1} of the standard normal distribution, i.e. $z_i = \Phi^{-1}(1 - p_i)$. An overall z-score can be derived as $z = \sum_i z_i / \sqrt{n}$, where n is the total number of p-values (i.e., the total number of aberrant chromosome arms, genomic regions, or genomic data modalities, depending on the context). If weights are available for each p-value (for example, if our confidence in each data modality is not the same), then a weighted overall z-score can be readily derived as $z = \sum_i w_i z_i / \sqrt{\sum_i w_i^2}$. In both cases, the overall z-score is distributed according to the standard normal distribution, from which an overall two-tailed p-value can be derived as $p = 2(1 - \Phi(z))$.

We adopted Stouffer's method because of its straightforward applicability to our problem, the ease of incorporating weights (if such weights become available), and the easy adaptability of the method to the absence of one or more data modalities from one or more patients (which is not uncommon in real-life clinical scenarios). Other approaches are possible (see our response to point 4 below), such as the very popular Fisher's method. However, in this case, integration of weights leads to sampling distributions that are not convenient to work with.

We have updated the Methods section **Statistical analysis for cancer detection** to clarify these points.

3. It is not clear how to convert integrated p-values to integrated scores. Please explain.

Given a p-value, a z-score can be derived using the inverse cumulative distribution function of the standard normal distribution. Please, see our response to the previous point for details.

4. It would be great if the authors could demonstrate comparisons between the new algorithm and other published algorithms in the same patient cohort.

Following the reviewer's comment, we have examined four additional published algorithms; specifically, Fisher's method (a popular alternative to Stouffer's method, as noted earlier), as well as Logistic Regression, Random Forest, and Support Vector Machine, all three of which are popular methodologies in the statistics and machine learning community. Fisher's method was used as a drop-in replacement of Stouffer's method for integrating different data modalities. The remaining three methods were used for building binary classifiers, where the response and predictors were the clinical status (cancer/non-cancer) and modality-specific scores of each patient, respectively. We assessed the performance of these methods on our cohort using leave-one-out cross-validation and we concluded that the best performing methods were Stouffer's and Fisher's methods (both having an AUC of 83.5%), followed by Logistic Regression (AUC 81.3%), Random Forest (AUC 80.7%) and Support Vector Machine (AUC 66.3%). These additions are now outlined in the Results section **Integration of multiple genomic modalities for ctDNA detection**, in the Methods section **Statistical analysis for cancer detection** and in Supplementary Figure S5.

5. The authors showed potential use of the integrated ctDNA score for detection of MRD and monitoring of treatment responses. Although the ctDNA scores in some cases were in good agreement with clinical outcomes, the results were based on a few cases and limited timepoints of blood collection. Longitudinal analysis with more blood collection times is needed.

We investigated the potential of using the integrated ctDNA score for detection of post-operative MRD and treatment response tracking in 10 surgical patients with colorectal cancer, half of which also received adjuvant therapy. Follow-up times for these patients were up to 8 years. Overall, the presence or absence of ctDNA correlated strongly with adjuvant therapy or clinical outcome in 9 out of 10 patients. Although based on a small cohort, this analysis was sufficiently powered to detect a difference in the event-free survival between patients with detectable and non-detectable ctDNA in the post-treatment plasma sample (Fig. 6Ci and Cii; HR 8.2, 95% CI 1.3 – 53.1, two-sided log-ranked test p-value = 0.02).

However, we agree with the reviewer that analysing follow-up samples in a larger cohort of patients (which currently is not possible due to cost considerations) is needed to increase our confidence in these results; for example, to estimate the above hazard ratio with higher precision. Following the reviewer's comment, we have added a sentence at the end of the Results section **Multimodal ctDNA detection for post-operative MRD and adjuvant therapy response tracking in colorectal cancer without matched tumour** to highlight this point in the revised text.

6. Due to high cost, this new approach that applies WGS with 80 x coverage may not be suitable for future clinical applications. Pairing with germline samples will further increase the overall cost. Other low-cost options could be a better option.

For a proper analysis of somatic mutations, pairing each plasma sample with a matched germline is essential. Although publicly available datasets (e.g., dbSNP) could be used to filter out common polymorphisms from plasma samples, these apply mostly to people of European origin and the post-filtering data are still expected to be heavily “contaminated” with germline mutations in the absence of a matched germline sample, which would unavoidably bias any attempted somatic analysis.

Furthermore, deep sequencing is necessary to increase the sensitivity of detecting somatic mutations. For example, at 80x and a rather high ctDNA fraction of 10%, a clonal heterozygous mutation in a diploid genomic region is expected to be supported by 4 reads (as calculated using the binomial distribution). At 40x, the expected number of supporting reads drops to 2. We usually require at least 3 supporting reads to call a mutation, therefore halving the depth of coverage would cause a drop in the total number of confidently called mutations and, therefore, in the sensitivity of detection. This reduction in sensitivity with decreasing coverage depth could be reversed by focusing our analysis on large patient-specific compendia of mutations. However, matched tumour biopsies, which could be used for generating such compendia, are typically not available in an early detection clinical setting, which was the focus of this study.

However, we understand the reviewer’s concern that the high cost incurred by deep sequencing and matching germline samples could pose an obstacle in the wider adoption of this approach, particularly in resource-restricted clinical settings. For this reason, future research should focus on determining the minimum feasible depth of coverage for this type of approach using down-sampling experiments made possible by this deep WGS sequencing dataset, as well as on the comparison against potentially cheaper long-read sequencing technologies.

Following the reviewer’s comment, this point is now highlighted in the revised Discussion section, paragraph 10.

7. The study generated a huge amount of data. However, there is no data-sharing information. The sequencing data should be deposited in the public domain for downloading.

We pledge to share the original sequencing data on the European Genome-Phenome Archive (EGA) after acceptance of the paper and before publication. This includes whole-genome TAPS data from 214 samples (plasma and matched germline pairs from 91 non-cancer and cancer subjects, as well as several follow-up plasma samples from 10 patients with colorectal cancer).

8. The manuscript should provide supplementary data to support results. For example, mutational data from all detected mutations to filtered mutations along their allele frequencies and sequencing depth. Also, information on cancer-associated methylation fragments needs to be included as supplementary materials.

Following the reviewer’s comment, we have added Supplementary Figures S2 and S8, as well as Supplementary Table S1. In Supplementary Figure S2, we provide a comparison of the

number of somatic mutations, coverage depth of mutated loci and variant allele fractions before and after filtering across all plasma samples. In Supplementary Figure S8, we illustrate the genome-wide depth of coverage of germline and plasma samples across all patients. In Supplementary Table S1, we provide the hyper-methylated regions from TCGA on which we focused our fragment-based methylation analysis. We further note that the overall survival and the frequency of mutated genes across patients are presented in Supplementary Figures S1 and S3, respectively, while further examples of treatment responses and disease monitoring are given in Supplementary Figure S7.

9. Line 477: "For the purposes of this study the threshold for classification as a tumour fragment was set at 80% modification." What is the rationale to use the 80% cutoff?

This threshold was applied on fragments overlapping regions that appear hyper-methylated in cancer. This implies that to classify a fragment as originating from a tumour cell, a threshold close to 100% had to be chosen, which however had to be, at the same time, sufficiently lower than 100% to account for tumour heterogeneity. Thus, the rationale behind the 80% threshold was to improve sensitivity, allowing for fragments where not all CpG positions were converted (for example, if methylated bases were replaced during end-repair or the chemistry was not 100% efficient), whilst not sacrificing the specificity of the fragment-based approach by ensuring that only well-converted fragments were considered as originating from tumour cells (i.e., fragments with fewer than 5 CpGs must be fully converted; fragments with between 5 and 9 CpGs can have only one CpG unconverted; fragments with between 10 and 14 CpGs can have at most two CpGs unconverted, and so on).

The above point is now explained in the revised Methods section **Fragment-based methylation analysis**.

10. What does "non-coding missense mutations" in lines 192-193 mean? Only protein-coding regions have missense mutations.

We meant to say "non-COSMIC missense mutations", i.e., missense mutations not found in COSMIC. We thank the reviewer for spotting this error. We have now corrected the disputed lines in the revised text.

11. The current algorithm lacks the capacity to classify tissue of cancer origin, which is essential in the current standard of multi-cancer detection.

The initial aim of this study was to develop a binary classifier for deciding the presence or absence of cancer. However, in response to the reviewer's comment, we have developed a multi-class classifier, which integrates all three data modalities (copy number aberrations, somatic single nucleotide variants and indels, and methylation signals) for predicting cancer type (see last paragraph in the Methods section **Statistical analysis for cancer detection**, and last paragraph in the Results section **Integration of multiple genomic modalities for ctDNA detection**), and we validated its performance using leave-one-out cross-validation (Supplementary Figure S6). The multi-class classifier correctly identifies the cancer origin in 25/36 colorectal, 4/8 oesophageal, 3/4 ovarian, 2/6 pancreatic and 0/5 renal cancer cases, leading to an overall balanced classification accuracy of 72%. At the same time, we correctly

identify as negative 8/9 non-cancer controls and as positive 56/59 cancer cases resulting in 94.9% sensitivity and 88.8% specificity. It is interesting to observe that among the 377 hypermethylated TCGA regions used in our methylation analysis, only 4 were specific to renal cancer, which may explain our inability to correctly classify any of the 5 corresponding cases in our cohort. Notice that we did not consider two cases with breast cancer, because it is not possible to conduct cross validation with only two cases in any of the cancer groups.

The above analysis is limited by the fact that our methylation signatures are based on methylation array data (covering approximately only 1% of the human genome) for a small set of cancer types from TCGA. This data comes mainly from colorectal cases, as this is where the TCGA dataset was most complete. To perform more precise tissue-of-origin analysis across many different tumour types, we need to have access to a methylation atlas of normal tissue generated using whole-genome TAPS. So far, such methylation atlases (including the TCGA methylation array data) have only been generated using bisulphite sequencing and, importantly, public access to them is restricted, as they are at the centre of commercial interests from companies like Grail, Exact Sciences, and others. A TAPS-specific methylation atlas is in preparation by Chun-Xiao Song (a co-author of this study and one of the inventors of TAPS) and colleagues and it will hopefully become publicly available soon. This is expected to enhance the accuracy of TAPS-derived tissue-of-origin (TOO) prediction. However, the debate whether TOO prediction is useful in clinical practice or whether it leads to an increase in false positive cancer calls is still on-going.

Following the reviewer's comment, we highlight the above points in the last paragraph of the Results section **Integration of multiple genomic modalities for ctDNA detection**, in the last paragraph of the Methods section **Statistical analysis for cancer detection**, and in paragraph 8 of the Discussion section in the revised manuscript.

13. It is unclear why cancer patients experienced more passenger mutations. Since these mutations are not cancer-driven, cancer patients may have similar passenger mutation frequency as controls, which needs further clarification.

The reviewer raises a very interesting point. Unlike driver mutations, passenger mutations are defined as those that do not confer a selective growth advantage (i.e., increased fitness) to the cancer cell compared to its surrounding, but rather occur coincidentally with the acquisition of a driver mutation and proliferate along with it with each cell division. However, this does not mean that cancer and non-cancer cells are expected to have the same number of passenger mutations, for the following reason: the higher rate at which cancer cells divide implies that passenger mutations accumulate at a higher rate, compared to non-cancer cells, even if we make the simplifying assumption that the probability at which a passenger mutation can occur at each cell division is the same in both cancer and non-cancer cells. In practice, this probability is expected to be higher in cancer cells, since mutation repair mechanisms are usually aberrant in cancer, which makes the accumulation of (passenger) mutations even more rapid, compared to non-cancer cells. In other words, the more often a cell divides, the more opportunities there are for a mutation to occur, and this leads to a higher number of passenger mutations in cancer cells, which is what we observe in our data.

We have added a note in the last sentence of the penultimate paragraph of the Results section **Analysis of somatic mutation burden for ctDNA** detection to clarify this point in the revised manuscript.

14. Please clarify why the study uses 21 non-cancer controls for data normalization and filtering but uses 9 non-cancer controls for case-control classification. Are the 9 controls from the 21 controls?

This study uses two groups of non-cancer samples. The first group includes samples from 21 non-cancer patients from the Oxford SCAN (Ssuspected CANcer) pathway. These patients presented to primary care with non-specific symptoms but did not develop cancer. The second group includes samples from 9 non-cancer subjects obtained from the company Cambridge Biosciences (CBS). The SCAN group was used for denoising, i.e., for removing non-cancer-specific unwanted variation from all other cancer and non-cancer samples, while the CBS group was used as a negative control group for case-control classification. Therefore, the 9 CBS samples were not included in the group of 21 SCAN controls. This is indicated in Figure 1 and throughout the revised text.

12. Lines 177-182 described 4 filtering criteria including d) further denoised moving all variants shared with any of 21 age-matched non-cancer plasma samples from patients. It is unclear if the somatic mutations reported in the paragraph starting from line 184 also went through the filtering criteria. If yes, the control plasma should have 0 somatic mutations. However, it reported 125-192 mutations in control samples.

Indeed, the somatic mutations reported in this paragraph are those that remained after the filtering process. The control plasma samples mentioned in this paragraph (and labelled as CTRL in the figures) are the 9 CBS non-cancer samples, which are separate from the 21 SCAN non-cancer samples that were used for denoising (as explained in our response to the previous point). Therefore, it is not surprising that the reported number of mutations in these 9 non-cancer samples is non-zero. The origin of all non-cancer samples is now clearly indicated throughout the main text to avoid confusion.

RESPONSE TO REVIEWERS' COMMENTS

Reviewer 3: expertise in ctDNA bioinformatics

Thank the authors for their effort in revising the manuscript. The new version of the manuscript has addressed most of my concerns and comments.

We would like to thank the reviewer for their time reading our paper and for their valuable comments. We are delighted that their concerns have been mostly addressed.

1) Although not required, I am wondering if the technology and algorithm can capture cancer (and cancer cell type) signals by downsizing sequencing coverage from 80x to 10x or even 1x. I think that the high cost of current 80x coverage is a limiting factor for further application of this technology in clinic.

This is indeed a sensible comment. We are currently conducting these down-sampling experiments, as well as alternative approaches based on shallow whole-genome long-read sequencing technologies. In response to the reviewer's comment, we have added the following paragraph in the discussion:

Furthermore, deep sequencing (at 80x) of both plasma and matching germline samples could pose an obstacle in the wider adoption of this approach, particularly in resource-restricted clinical settings. For this reason, future research should also focus on determining the minimum feasible depth of coverage for this type of study using down-sampling experiments made possible by the deep whole-genome sequencing dataset we present here, as well as on the comparison against potentially cheaper long-read sequencing technologies.

2) I am also wonder if authors can convert the TAPS-based methylation to bisulfite-based methylation format. In doing so, authors can use existing deconvolution tool for cell type deconvolution.

This should be possible, in principle. However, we hope that the imminent release of a whole-genome TAPS-based methylation atlas will drive the adaptation of these deconvolution tools to this type of methylation format, in addition to bisulphite-based data. In response to the reviewer's comment, we have added the following text to the discussion:

To perform more precise tissue-of-origin analysis across many different tumour types, we need to have access to a methylation atlas of normal tissue generated using whole-genome TAPS. So far, such methylation atlases (including the TCGA methylation array data) have only been generated using bisulphite sequencing. A TAPS-specific methylation atlas is in preparation (Chun-Xiao Song, personal communication) to enhance the accuracy of TAPS-derived tissue-of-origin (TOO) prediction.